# Regulation and Function of Tumor-Associated Macrophages (TAMs) in Colorectal Cancer (CRC): The Role of the SRIF System in Macrophage Regulation

**DOI:** 10.3390/ijms26115336

**Published:** 2025-06-01

**Authors:** Agnieszka Geltz, Jakub Geltz, Aldona Kasprzak

**Affiliations:** 1Department of Histology and Embryology, Poznan University of Medical Sciences, Swiecicki Street 6, 60-781 Poznan, Poland; 72784@student.ump.edu.pl; 2Doctoral School, Poznan University of Medical Sciences, Bukowska Street 70, 60-812 Poznan, Poland; 75159@student.ump.edu.pl; 3Department of Pediatric Gastroenterology and Metabolic Diseases, Poznan University of Medical Sciences, Szpitalna Street 27/33, 60-572 Poznan, Poland

**Keywords:** colorectal cancer, macrophages, polarization, tumor-associated macrophages, somatostatin, somatostatin receptors, role of SRIF system in immune system

## Abstract

Colorectal cancer (CRC) remains the leading cause of morbidity and mortality for both men and women worldwide. Tumor-associated macrophages (TAMs) are the most abundant immune cells in the tumor microenvironment (TME) of solid tumors, including CRC. These macrophages are found in the pro-inflammatory M1 and anti-inflammatory M2 forms, with the latter increasingly recognized for its tumor-promoting phenotypes. Many signaling molecules and pathways, including AMPK, EGFR, STAT3/6, mTOR, NF-κB, MAPK/ERK, and HIFs, are involved in regulating TAM polarization. Consequently, researchers are investigating several potential predictive and prognostic markers, and novel TAM-based therapeutic targets, especially in combination therapies for CRC. Macrophages of the gastrointestinal tract, including the normal colon and rectum, produce growth hormone-releasing inhibitory peptide/somatostatin (SRIF/SST) and five SST receptors (SSTRs, SST1-5). While the immunosuppressive function of the SRIF system is primarily known for various tissues, its role within CRC-associated TAMs remains underexplored. This review focuses on the following three aspects of TAMs: first, the role of macrophages in the normal colon and rectum within the broader context of macrophage biology; second, the various bioactive factors and signaling pathways associated with TAM function, along with potential strategies targeting TAMs in CRC; and third, the interaction between the SRIF system and macrophages in both normal tissues and the CRC microenvironment.

## 1. Introduction

Colorectal cancer (CRC) is one of the most common human malignancies worldwide, ranking third in incidence and second in mortality [1]. Notably, approximately 50% of patients develop distant metastases, with the liver being the most common site. CRC occurs primarily as colon adenocarcinoma and affects adults of all age groups. A study spanning 50 countries and territories revealed an increasing trend in CRC incidence in younger adults (aged 25–49 years) compared to older adults (aged 50–74 years) in 27 of those countries [2,3].

The tumor microenvironment (TME) is involved in the activation and recruitment of inflammatory cells, angiogenesis, and extracellular matrix remodeling, leading to tumor progression [4,5]. Among the immune cells in the TME of CRC, macrophages are the most abundant population of inflammatory cells [6,7]. They are then called tumor-associated macrophages (TAMs) and exhibit a dualistic nature, functioning as either tumor suppressors (M1 phenotype) or promoters (M2 phenotype), depending on their intricate polarization state. This inherent plasticity highlights the need for a deeper understanding of these mechanisms to facilitate the development of targeted TAM therapies [8]. The presence of a high number of TAMs with a CD68+ phenotype and infiltrating CD8+ cytotoxic lymphocytes has led to the identification of two immunogenic CRC subtypes, namely, consensus molecular subtype 1 (CMS1) and CMS4 [9].

The development of CRC is a complex interplay among genetic, epigenetic, and environmental factors. Extensive research has elucidated the major signaling pathways, various neuroendocrine (NE) peptides, and growth factors involved in colon carcinogenesis [10,11,12,13,14]. Many of these signaling molecules, including epidermal growth factor (EGF), fibroblast growth factor 2 (FGF-2), insulin-like growth factor 1 (IGF-1), vascular endothelial growth factor (VEGF), and transforming growth factor β (TGF-β), exert their pro-cancer mechanisms of action and affect the regulation of TAMs [6,7].

The growth hormone-releasing inhibiting peptide/somatostatin (SST/SRIF) system is an interesting system involved in the interactions between tumor cells and other TME cells in CRC. This system consists of seven genes encoding two peptide precursors, namely SST and cortistatin (CST), and five receptor genes (SST1-5) [15]. Notably, studies show that macrophages themselves also produce SST, which is considered one of the immunoregulatory cytokines in inflammation [16,17,18,19,20]. The main receptor type mediating the autocrine and paracrine effects of SST in immune cells, including macrophages, is SST2 [19,21,22,23]. This results in reduced phagocytic activity of human monocytes and macrophages [18,22,24]. While attempts have been made to use TAMs as a target for SST analog (SSA) therapy in the treatment of inflammation [25] and some human malignancies [26,27,28] with varying degrees of success, the role of TAMs as a target for SST system components in CRC therapy remains an open question.

This review focuses on three key aspects of TAMs in CRC. First, we summarize current knowledge regarding the role of macrophages in the normal colon and rectum, in the context of the broader understanding of macrophage biology. Second, we present a comprehensive overview of the various bioactive factors and signaling pathways associated with TAM function and potential strategies for targeting TAMs in CRC. Finally, we discuss the role of the SRIF system and its interactions with macrophages in normal tissues and in the CRC microenvironment.

## 2. Characteristics and Nomenclature of Macrophages

Macrophages are present in all tissues/compartments of the body [29,30,31,32,33]. Interest in these cells developed following the discovery of phagocytosis by the father of cellular immunology, Èlie Metchnikoff (1845–1916). He described the role of “eating” cells, i.e., phagocytes and microphages (now neutrophils and polymorphonuclear leukocytes), in response to injury, inflammation, and infection [34]. Macrophages would be expected to play a role in maintaining tissue integrity and homeostasis [31,35]. Metchnikoff also pointed out the link between macrophages and the microbiome in the mechanisms of aging. He also predicted the existence of the so-called bacterial translocation and linked chronic inflammation to the pathogenesis of many diseases (including cancer) [34].

### 2.1. The Origin of Macrophages

For almost half a century, the prevailing view has been that tissue-resident macrophages (TRMs) are representatives of the human mononuclear phagocyte system (MPS) and are derived from circulating monocytes [36]. However, the conventional MPS theory has been challenged by experimental evidence in mice and human organ transplant studies, which has been described in detail in other reviews [33,37,38]. There is an ongoing debate over whether embryonic and monocyte-derived TRMs respond similarly to disease signals. Existing studies suggest that the ontogeny of these cells is important [39]. In mice, macrophages localize in multiple organs; originate from precursors in the yolk sac, liver, and spleen of the fetus; and persist as TRMs throughout adult life. In other tissues, such as the gastrointestinal (GI) tract, TRMs are formed from monocyte precursors. In many adult mouse tissues, the population of TRMs consists of a mixture of cells originating during development and those derived from circulating monocytic precursors [37,40].

The dual origin of tissue macrophages also applies to human mononuclear phagocytes [41]. With the development of transcriptomics and high-resolution imaging tools, more subpopulations of TRMs with different functions are being discovered. Thus, it is now believed that TRMs originate from embryonic precursors and are seeded prenatally in tissues and maintained by self-renewal throughout adulthood [42].

### 2.2. Classification and Morphological Characteristics of Macrophages

Most commonly, tissue macrophages consist of a mixed population of resident macrophages of embryonic origin and marrow-derived circulating monocytes (migrating macrophages). Both TRMs and monocyte subpopulations in the blood show phenotypic differences that reflect the heterogeneity associated with their origin, maturation, and activation (reviewed in [43]). Researchers also found that the factors shaping the phenotypes and functions of TRMs can be organized into four main points, as follows: ontogeny, local environment, inflammation, and the time of residence in tissues [38].

Tissue macrophages were described by Metchnikoff as “wandering, amoeboid cells” [35]. The macrophage’s high endocytic ability can be seen in its irregular shape and high number of extensions of the cell membrane, which allow it to catch extracellular material quickly [44]. Once mature, they assume different shapes, depending on their function, and are usually larger than monocytes. They are characterized by the presence of numerous surface and cellular markers, sometimes shared with dendritic cells (DCs) and other immune cells [33]. Wandering macrophages are also larger than TRMs. They are characterized by a variable shape, again resembling an amoeba with short, blunt spears. TRMs have long and narrow spears and are characterized by negligible motility. Their nuclei are round or oval. The cytoplasm is acidophilic and poor in cell organelles, except for lysosomes, which are abundant [43]. Mills et al. proposed a classification of M1 and M2 macrophages based on observations of the occurrence of different metabolic programs. Macrophage TGF-β1, which inhibits inducible nitric oxide synthase (iNOS) and stimulates arginase, is thought to play a role in regulating the balance between M1 and M2 macrophages [45]. Later, Mantovani et al. subdivided M2 macrophages into M2a, M2b, and M2c [46]. Depending on the stimuli used and the resulting transcriptional changes, subpopulations M2a, M2b, M2c, and M2d are finally distinguished among M2 macrophages [47,48,49,50]. Morphometric studies indicate that M1 macrophages are larger and rounder, while M2 macrophages are more elongated [51,52]. Cell elongation alone, without exogenous cytokines, leads to the expression of M2 phenotype markers and reduces the secretion of pro-inflammatory cytokines. It also enhances the effect of M2-inducing cytokines (e.g., interleukin-4 (IL-4) and IL-13) and protects cells from M1-inducing stimuli, i.e., lipopolysaccharide (LPS) and interferon-gamma (IFN-γ). A role for the cell cytoskeleton in the control of macrophage polarization has also been suggested [51]. The effects of various agents (including chemotherapeutics) have been shown to alter macrophage polarization in vitro [52,53]. It has also been suggested that the M1/M2 paradigm is a vast oversimplification, and that TRMs are much more complex cells with a full range of identities and activation states [37,38].

Selig et al. have confirmed the existence of up to six different phenotypes of human macrophages and their IL-10 content based on single-cell morphology. Depending on the microenvironmental stimuli and activation state, these authors distinguished the M0, M1, and M2 (M2a and M2c) subpopulations of macrophages. Thus, M0 cells and granulocyte-macrophage colony-stimulating factor/tumor necrosis factor-alpha (GM-CSF/TNF-α)/IFN-γ-M1 cells expressed the highest intracellular IL-10 intensity, followed by macrophage colony-stimulating factor (M-CSF)/IL-10-M2c macrophages. In contrast, M-CSF/IL-4-M2a macrophages expressed the lowest IL-10 intensity [54]. Plaque-specific macrophage phenotypes (e.g., Mox, Mhem, and M4) were also identified [48].

### 2.3. General Functions of Macrophages

Classic adaptive responses of macrophages include tolerance, priming, and a broad spectrum of activation states, including M1, M2, and M2-like states [29,30,33]. Undifferentiated macrophages (M0) can be polarized into two types, namely, classically activated macrophages (M1) and alternatively activated macrophages (M2) [48,50,54]. M1 and M2 macrophages influence different types of inflammatory responses. M1 macrophages exhibit pro-inflammatory functions, whereas M2 macrophages have anti-inflammatory functions and promote immune regulation and tissue remodeling [47,55]. Beyond infection, M2 macrophages play a role in the resolution of inflammation through high endocytic clearance capacity and synthesis of trophic factors, accompanied by reduced secretion of pro-inflammatory cytokines [47,55]. M2 macrophages are polarized by Th2 cytokines, such as IL-4 and IL-13, and produce anti-inflammatory cytokines, e.g., IL-10 and TGF-β [56]. The phenotypic heterogeneity of TRMs is considered to be a critical determinant of the immune response [41,56,57,58] and cell behavior [43]. Little is known about macrophage heterogeneity and plasticity in humans [41,59]. Moreover, the range of functional states of macrophages seems debatable [60]. It has been argued that even modern molecular biology techniques do not always allow for a complete and proper assessment of the subpopulations of these cells. Heterogeneity may, therefore, be an adaptive feature of innate immunity, ensuring that each macrophage presents a unique challenge to potential pathogens [57]. The high plasticity of TRMs allows them to perform many homeostatic functions. Thus, they can be recognized as a plenary component of tissue, not just as immune cells responsible for defense against pathogens [38].

There are multiple mechanisms of macrophage polarity in health and disease (reviewed in [61,62]). M2a and M2b macrophages play an immunomodulatory role and promote Th2 cell responses, while M2c macrophages are associated with immune response and tissue remodeling. Additionally, the concept of M2d macrophages has emerged, that is, a subset activated by Toll-like receptors, specifically characterized by the expression of VEGF, TGF-β, and IL-10. These macrophages play a role in angiogenesis and tumor progression, and often only this subpopulation is referred to as TAMs [48,50,56,63,64]. By comparing macrophages at the tumor site and the maternal-fetal interface, they generally adopt a homeostatic phenotype similar to M2, resulting in negative outcomes (tumor progression and spreading) and positive outcomes (fetus development and pregnancy maintenance), respectively [59]. Noteworthy, macrophages may also play a role in tissue organogenesis and morphogenesis [31].

## 3. Colon and Rectal Macrophages in Physiology

Metchnikoff proposed the involvement of interactions between macrophages and the colonic microbiome in the aging process. In his opinion, phagocytes, transitioning from defenders against infections, could contribute to age-related disorders due to autotoxins produced by putrefactive colonic bacteria, leading to the destruction of healthy tissues [34].

Macrophages are present in varying numbers in the lamina propria (Lp) mucosae (LpM) throughout the human GI tract, with the highest density in the colon and rectum. They possess typical markers, such as CD68, laminin 5 (LN5), lysozyme, ferritin, and alpha 1-antichymotrypsin, and are mainly involved in the regulation of epithelial cell renewal. Other expressed markers include 25F9 (a marker for a certain subpopulation of macrophages), major histocompatibility complex proteins class II (MHC-II), and CD74 (MHC-II-associated invariant chain, M1 marker) [65]. With the advancements in single-cell transcriptomics, tremendous heterogeneity within the macrophage pool has been revealed, suggesting niche-specific functional specialization of TRMs [66,67]. Two macrophage subsets have been identified in the human colon, that is, acid phosphatase 5 (ACP5)+ complement component 1Q (C1Q)+ macrophages, known as Lp macrophages (LpM), and lymphatic vessel endothelial hyaluronan receptor 1 (LYVE1)+ collectin subfamily member 12 (COLEC12)+ macrophages located in the colonic submucosa. LpMs include a subgroup with pro-inflammatory properties and a subgroup with a high capacity for antigen presentation and phagocytosis. The latter are strategically located close to the surface epithelium. There are also muscularis macrophages, most of which differentiate through two signaling pathways, that is, one that enhances the regulation of genes related to immune activation and angiogenesis, and a second that upregulates genes related to neuronal homeostasis [67]. In addition to their key role in pathogen clearance, colon macrophages also regulate inflammatory responses, local tissue homeostasis, and insulin sensitivity. The term “meta-inflammation” has been introduced to describe chronic inflammation caused by macrophages located in the colon, liver, muscle, and adipose tissue (reviewed in [61]).

In the mouse large intestine, macrophages are also located in the Lp, directly adjacent to the epithelial crypt cells [68]. The role of these subepithelial LpMs in gut homeostasis, particularly through their interactions with the microbiota, has been described [43,69]. This mechanism is based on a population of distal colonic macrophages, equipped with epithelium-anchored “balloon-like” protrusions (BLP) that absorb fluids. This allows for rapid control of the quality of absorbed fluids to prevent colonocyte poisoning by fungal toxins. In the absence of macrophages or BLP, epithelial cells continue to absorb fluids that contain these toxic products, leading to their death and subsequent loss of epithelial barrier integrity [69]. A recent comprehensive study in a mouse model describes a novel mechanism by which colonic macrophages deliver polyamines (spermidine and spermine) to epithelial cells to support epithelial self-renewal during periods of high proliferation [68,70]. Macrophages have been likened to “commensals” that provide metabolic support to promote efficient self-renewal of the colonic epithelium [68].

Nerve fibers were more frequently found in contact with macrophages in the cecum than in the descending colon in rats [71]. Subcellular differences in macrophages were demonstrated, depending on the anatomical location of the rat colon. Macrophages in the cecum had a higher number of lysosomes and a lower number of vacuoles than macrophages in the descending colon. Macrophages with many intraepithelial cell extensions were observed beneath the intestinal epithelium in the descending colon. Furthermore, macrophages in contact with nerve fibers were more common in the cecum than in the descending colon, and their subgroup surrounded nerve bundles only in the cecum [72].

## 4. Characteristics of Tumor-Associated Macrophages (TAMs) in CRC

TAMs, classified as tumor-infiltrating immune cells [73], are among the most abundant immune cells in the TME of CRC [6,7]. Numerous studies suggest an ambivalent role of TAMs in the progression and development of CRC [6,74,75,76]. The exact role of these cells in human CRC remains unclear [7,61,77,78,79,80,81].

Compared to normal mucosal macrophages, CRC macrophages have an altered phenotype, probably due to defective maturation and polarization in the tumor, resulting in a more pro-inflammatory state [77]. Two sets of TAMs have been distinguished in CRC, which express different transcription factors, that is, avian musculoaponeurotic fibrosarcoma protooncogene/MAF homolog B (MAF/MAFB) and proto-oncogene C-Fos/Jun proto-oncogene (FOS/JUN) in complement C1q C chain (C1QC)+ TAMs, and CCAAT enhancer binding protein beta (CEBPB) and zinc finger E-box binding homeobox 2 (ZEB2) in secreted phosphoprotein 1 (SPP1)+ TAMs. Single-cell RNA sequencing analyses revealed two distinct TAM subsets with inflammatory and angiogenic characteristics, respectively. They also had different sensitivity to colony-stimulating factor 1 receptor (CSF-1R) blockade. Color imaging studies confirmed the presence of two TAM subsets through co-expression of (1) CD68, CD80, and MAF molecules and (2) CD68, macrophage receptor with collagenous structure (MARCO), and VEGFA molecules, respectively [82].

Recent studies indicate unique human macrophage niches that function as basic building blocks in colon tissues and may influence tumor growth. Their findings distinguish the NLR family pyrin domain containing 3 (NLRP3)+ and SPP1+ macrophages as associated with the TME from folate receptor beta 2 (FOLR2)+ and LYVE1+ macrophages found in normal tissues. A marker of malignant macrophages in CRC is interleukin-4-induced gene 1 (IL-4I1), although its presence has also been observed in the normal colon [83].

Studies suggest that macrophage infiltration in CRC tissues also increases with age. Macrophages from older individuals were more susceptible to polarization toward a pro-tumor phenotype and more strongly promoted cancer cell proliferation [84]. Ugai et al. reported a tendency toward a lower density of total macrophages and M1-like macrophages in intermediate-onset CRC (50–54 years) than in late-onset CRC (≥50 years) [5].

Metastasis-associated macrophages (MAMs) have also been identified. In mouse models of breast cancer and CRC, MAMs have been shown to increase tumor cell extravasation and promote survival by secreting growth factors and simultaneously inhibiting cytotoxic T lymphocytes [85]. CRC liver metastases (CRLMs) are highly heterogeneous, accounting for a wide range of clinical symptoms and treatment responses (reviewed in [86]). Geng et al. demonstrated a significantly increased number of M2 macrophages (TAMs2) in CRLM compared to primary tumors, where there were fewer myeloid cells, and they were mainly monocytes and TAMs1. High expression was seen for the myeloid-derived suppressor cell (MDSC) characteristic gene and mannose receptor C-type 1 (MRC1). The TME in CRLM is, therefore, significantly immunosuppressive. Interactions between myeloid cells and other TME cell populations contribute to the formation of a pro-metastatic niche that promotes CRC cell colonization and growth in the liver [87]. The detailed mechanisms by which TAMs promote the CRC metastatic cascade have been the subject of a recent excellent review [88].

### 4.1. Regulatory Factors of TAMs in CRC

The interactions between TAMs and cancer cells (including CRC cells) are regulated by numerous soluble factors [88,89,90,91], surface-bound molecules [92], and physical cues [51]. The polarization of M1 macrophages is induced by soluble factors such as LPS, TNF-α, and IFN-γ, while the M2 macrophage population is induced by cytokines such as IL-4, IL-10, and IL-13 [88,93,94]. Notably, CRC cells themselves are capable of producing soluble factors that influence monocyte differentiation, often resulting in the induction of suppressive subgroups [89,93]. Different studies observed the subsequent induction of mainly tumor-promoting M2 macrophages [62,93,95,96,97]. An increasing number of signaling molecules have been implicated in the modulation of TAM polarization, functioning within many of the established signaling pathways involved in colorectal carcinogenesis. These include a wide range of proteins (acting as transcription factors, translation factors, enzymes, and oncoproteins), extracellular vesicles (EVs), epigenetic factors (non-coding RNAs), components of the colonic microbiota, and natural bioactive compounds, which will be presented in the following subsections of the review.

#### 4.1.1. Selected Proteins and Extracellular Vesicles (EVs)

The selected proteins (including oncoproteins) and EVs affecting the regulation of TAMs in CRC presented in this section are summarized according to their involvement in a signaling pathway with a recognized role in colorectal carcinogenesis. One of the factors that has been identified as contributing to the regulation of colonic macrophages in the early stages of colorectal tumorigenesis is EGFR, which is considered the “prototypical” receptor tyrosine kinase [98]. Inhibition of the EGFR secreted by CRC cells modulated the expression of IGF-1 and prevented the polarization of M1 to M2 macrophages, resulting in the inhibition of tumor cell growth [98]. Hardbower et al. have demonstrated the involvement of EGFR signaling in macrophages themselves, but not in CRC epithelial cells, in the promotion of colitis-associated CRC (CAC) in a mouse model. In the myeloid-specific *Egfr*-deficient mouse model, a reduction in both M2 and M1 macrophages was demonstrated. This resulted in a decrease in the production of proangiogenic factors and neoangiogenesis. Strong EGFR activation in colonic macrophages was observed in human tissue microarrays (TMAs) during colitis and dysplastic changes [99]. Deng et al. demonstrated the role of cannabinoid receptor-1 (CB1) activation in inhibiting M2 macrophage differentiation through EGFR downregulation. This, in turn, resulted in the inhibition of CRC cell proliferation, migration, and invasion [100].

The induction of changes in macrophage polarization (a decrease in M2 and an increase in M1 macrophage population), the regulation of epithelial–mesenchymal transition (EMT) of CRC cells, and activation of phosphoinositide 3-kinase/serine/threonine kinase Akt (PI3K/AKT) signaling are also affected by proprotein convertase subtilisin/kexin type 9 (PCSK9). This enzyme, which plays a key role in the regulation of lipoprotein homeostasis, was found to directly or indirectly upregulate the expression of Snail1, leading to decreased E-cadherin expression and increased levels of N-cadherin and matrix metalloproteinase-9 (MMP9). Only PCSK9 knockout inhibited M2 macrophage migration and promoted M1 macrophage polarization, an effect mediated through macrophage migration inhibitory factor (MIF) and altered lactate levels [101].

As demonstrated, the process of macrophage polarization toward the M2 phenotype is also subject to regulation by the malignant fibrous histiocytoma amplified sequence 1 (MFHAS1) oncoprotein. This regulatory process occurs through the upregulation of two transcription factors, i.e., signal transducer and activator of transcription 6 (STAT6) and Krüppel-like factor 4 (KLF4) [102].

In interactions between CRC cells and TAMs involving the IL-6R/STAT3 signaling pathway, the role of cytoplasmic polyadenylation element binding protein 3 (CPEB3)—a highly conserved RNA-binding protein—was also confirmed. Decreased expression of CPEB3 resulted in chemokine (C-C motif) ligand 2 (CCL2)-induced M2-like TAM polarization and IL-6-induced EMT in CRC cells [103]. Increased expression of plexin domain containing 1 (PLXDC1, tumor endothelial marker 7, TEM7) was also observed in CRC tissues. The expression of this protein was shown to correlate with markers of M2 macrophages. IL-6/STAT3 signaling has also been implicated in the mechanism of action [104]. High-mobility gene group A2 (HMGA2) enhanced macrophage recruitment and polarization toward the M2 phenotype. This oncoprotein acts in the HMGA2/STAT3/CCL2 signaling pathway both in vitro and in vivo [105].

Mammalian target of rapamycin (mTOR) and nuclear factor kappa-light-chain-enhancer of activated B cells (NF-κB) have also been implicated in changing the phenotype of TAMs. Li et al. demonstrated that metastasis-related secretory protein cathepsin K (CTSK) produced by CRC cells binds to TLR4 and promotes M2 macrophage polarization through an mTOR-dependent pathway. Furthermore, they found that tumor invasion and metastasis are mediated through activation of the NF-κB pathway [106].

Metformin (MET)—an adenosine monophosphate-activated protein kinase (AMPK) agonist—also shows a modulating effect on M2 macrophage polarization [107,108]. In the conversion of M2 to M1, the role of MET was to inhibit the hypoxia-inducible factor 1 alpha (HIF-1α) and mTOR signaling pathways [107]. In the enterotoxigenic *Bacteroides fragilis* (ETBF)-associated CRC mouse model, modulation of macrophage polarization and inhibition of CRC progression by MET were observed through the mechanism of suppression of the TLR4/MyD88/NF-κB/MAPK pathway. Furthermore, a protective role of MET was observed by increasing short-chain fatty acid (SCFA) levels in the colon [108]. Both of these established mechanisms of MET action resulted in the inhibition of colon carcinogenesis.

Protein kinase 2 (PKN2), a protein kinase C (PKC)-related serine/threonine-protein kinase, is also known to be involved in altering the phenotype of macrophages in CRC in vitro and in vivo. From a mechanistic perspective, PKN2 inhibited the expression of IL-4 and IL-10 in CRC cells by suppressing dual specificity phosphatase 6/extracellular signal-regulated kinase 1/2 (DUSP6/ERK1/2) signaling [109].

Additionally, observations suggest the involvement of extracellular vesicles (EVs)—membrane-bound vesicles derived from primary and metastatic CRC cell lines (SW480 and SW620, respectively)—in reprogramming the immunophenotype and secretory profile in the THP-1 line monocyte differentiation model. EVs from primary CRC cells modulated the immunophenotype and secretory profile of monocytes and quiescent macrophages toward an M1-type response. In contrast, EVs from metastatic CRC cells induced a mixed M1 and M2 cytokine response in quiescent macrophages. The exposure of monocytes and M0 macrophages to CRC cell EVs may contribute to the migration of monocytes toward the TME and the induction of a pro-inflammatory response by TAMs [110]. Another novelty is the use of M1 macrophage-derived EVs (M1EVs) as nanocarriers of oxaliplatin (OXA) (M1EV1) associated with retinoic acid (M1EV2) and *Libidibia ferrea* (Brazilian ironwood) (M1EV3), either alone or in combination (M1EV4). The use of this therapy in CRC resulted in a change from the M2 phenotype (CD163+) to the M1 phenotype (CD68+), with a reduction in the levels of IL-10, TGF-β, and CCL22 [111].

The involvement of functionally different proteins and EVs in colorectal carcinogenesis through TAM polarity changes is shown in Table 1.

#### 4.1.2. Non-Coding RNAs (ncRNAs)

Non-coding RNAs (ncRNAs) are involved in TME cell interactions and immune response in CRC (reviewed in [114]). Most of them are dysregulated in CRC, which has prognostic significance [114,115,116,117]. It has been shown that ncRNAs are directly or indirectly involved in the regulation of TAMs in CRC [117,118,119]. The ncRNAs include small ncRNAs that are less than 200 nucleotides in length (e.g., microRNAs (miRNAs), small interfering RNAs, Piwi-interacting RNAs, small nuclear RNAs, and small circular RNAs) and long non-coding RNAs (lncRNAs) greater than 200 nucleotides in length. Although they do not encode proteins, they play a key role in tumor progression and are ectopically expressed in malignant tumors [120,121]. It is worth mentioning that miRNAs, circRNAs, and lncRNAs are also clearly visible in exosomes. Exosomes are part of EVs, which act as channels for information exchange between cells and come from many sources. Exosome-derived ncRNAs play an important role in tumor development, progression, immune evasion, and drug resistance, including in CRC (reviewed in [117]). Many of them are involved in pro-tumor polarization of macrophages (M2) in CRC [118,122,123,124,125,126,127].

Most lncRNAs promote CRC progression and are involved in metastasis and/or drug resistance. This occurs through different mechanisms and signaling pathways. In most cases, there is an increase in the polarization of TAMs toward the M2 phenotype. This is caused by the following lncRNAs, namely lncRPPH1 [118], lncPTTG3P [128], lncHLA-F-AS1 [129], lncMIR155HG [130], lncHOXB8-1:2 [122], lncHCG18 [131], LINC00543 [132], lncXIST [124], lncRP11-417E7.1 [125], and lncBANCR [126]. The miRNAs directly responsible for the increase in TAM2 polarization include miR-21-5p and miR-200a [133], miR-106a-5p [127], miR-122 [134], and circ-0034880 [135]. LncNBR2 [136,137] is thought to be responsible for the decrease in TAMs2 and the simultaneous increase in TAMs1.

The decrease in TAMs2 is also caused by miR-216b [138], miR-4766 [139], and miR-I48a [140], and hUC-MSCs-Exos carrying miR-1827 [141]. And the increase in TAM1 polarization, in addition to lncNBR2 [136,137], is caused by miR-18a encapsulated in grapefruit-derived nanovector (GNV) [142]. The use of optimized-GNV (OGNV) led to an increase in the percentage of F4/80+MHC-II+, F4/80+IL-12+, F4/80+IFNγ+, and F4/80+CD80+ cells. This type of miRNA induced macrophage IFN-γ by targeting interferon regulatory factor 2 (IRF2), which is required for subsequent IL-12 induction. This cytokine activated natural killer (NK) and natural killer T (NKT) cells to inhibit CRC liver metastasis [142].

Some molecular mechanisms in CRC metastasis have also been revealed through which the gut microbiota cooperates with miRNAs. Among others, *Porphyromonas* and *Bifidobacterium* spp. were shown to be associated with the majority of miRNAs, and has-miR-3943 was the target of most types of microorganisms. Moreover, five types of immune cells (including M1 macrophages) were shown to differ between the groups with metastases (M1 stage) and those without (M0 stage). The relative numbers of infiltrating activated NK cells, M1 macrophages, and resting mast cells were significantly higher in the M0 group than in the M1 stage group [143].

The involvement of selected ncRNAs cooperating with TAMs in CRC progression and drug resistance is shown in Table 2.

#### 4.1.3. Components of the Colonic Microbiota and Natural Bioactive Compounds

The role of the tumor-resident microbiota in modulating tumor immunity remains unclear. However, some bacteria or their metabolites may influence TAM polarization in CRC [97,144]. In most cases, bacterial involvement in TAM reprogramming occurs through well-known signaling pathways, e.g., NF-κB [145,146,147,148]. Colon microbiota often interact with selected ncRNAs [146,148]. For example, *Akkermansia muciniphila* (*A. muciniphila*), a SCFA-producing bacterium, promotes the expression of M1-like macrophages and has been suggested as a potential target in CRC therapy [145]. However, closer analysis suggests that the role of *A. muciniphila* in inhibiting the development and treatment of human CRC remains controversial (reviewed in [149]). *Fusobacterium nucleatum* (*F. nucleatum)* promotes macrophage infiltration by activating CCL20 while inducing M2 macrophage polarization. The mechanism of pro-metastatic action of this bacterium would be related to the reduction of miR-1322 expression by activating the NF-κB signaling in CRC cells and M2 macrophage polarization [146]. Other studies confirmed the role of *F. nucleatum* in M2 macrophage polarization and CRC progression through the mechanism of TLR4/NF-κB/S100A9 signaling activation [147].

*F. nucleatum* autoinducer-2 (AI-2) was found to be an inhibitory factor in CRC progression [150,151]. Li et al. were the first to demonstrate elevated levels of AI-2 in the colon tissue and stool of CRC patients compared to adenoma and the normal colon. They also observed an increase in AI-2 levels with CRC progression. After AI-2 stimulation, the most significant increase in TNF superfamily member 9 (TNFSF9) protein expression was observed in macrophages. This means that the source of increased TNFSF9 expression in CRC tissues was mainly macrophages in the TME. Moreover, AI-2 levels were positively associated with the number of CD3+ T cells and negatively associated with the CD4/CD8 ratio in CRC tissues [151]. The same research team showed that *F. nucleatum* AI-2 increased macrophage mobility and M1 polarization via TNFSF9/TNF receptor-associated factor 1/phosphorylated AKT/IL-1β (TNFSF9/TRAF1/p-AKT/IL-1β) signaling [150]. In contrast, *Escherichia coli* (*E. coli*), which increases lactate production, mediates M2 macrophage polarization by suppressing NF-κB signaling through lactylation of retinoic acid-inducible gene 1 (RIG-I) [152].

Plant-derived factors used therapeutically in Chinese medicine, e.g., *Astragalus mongholicus* Bunge and *Curcuma aromatica* Salisb., used as a concoction, and abbreviated as ARCR, appear to inhibit CRC growth and metastasis by inhibiting the polarization of M2 macrophages. Some ncRNAs (e.g., miR-153-3p) are also involved in their action [148]. The F1 fraction of *Mesobuthus eupeus* has also been studied in the regulation of M2 to M1 macrophage polarization in CRC [153]. Scorpion venoms are known to have anti-tumor and anti-angiogenic properties, as shown in several cell lines (including CRC cells) (reviewed in [154]). Inhibition of CRC cell proliferation and increased apoptosis through increased reactive oxygen species (ROS) and mitochondrial depolarization have also been shown in another study [155]. The studies by Sadeghi et al. confirmed the involvement of this natural factor in the upregulation of cytokines and TAMs1 markers and the reduction of TAMs2 markers, thereby indicating its inhibitory effect on CRC carcinogenesis through this mechanism [153].

Inhibition of carcinogenesis in the CAC model (but not in non-chronic CAC) was also demonstrated after berberine (an isoquinoline alkaloid) treatment. A reduction in the number of infiltrating macrophages in the colon was observed in vivo. In vitro studies, on the other hand, showed a significant reduction in the percentage of M1 macrophages and the levels of cytokines (e.g., IL-1β, IL-6, and TNF-α). Additionally, berberine reduced miR-155-5p levels and increased the expression of suppressor of cytokine signaling 1 (SOCS1) in cells. These studies suggest that the inhibitory effect of berberine on the development of CAC is related to its anti-inflammatory activity. And the miR-155/SOCS1 pathway is required for berberine’s modulation of the carcinogenic process [156]. Natural bioactive compounds that reduce the number of M2 macrophages in CRC also include Cuban brown propolis (Cp) and its main component nemorosone (Nem). They reduced the viability of M2 macrophages and, thus, the activity of the matrix metalloproteinase 9 (MMP-9). The use of Cp/Nem prevented the production of soluble factors by M2 and inhibited the growth, migration, and invasion of CRC cells (HT-29 cell line) [90] (Table 3).

Figure 1 shows a schematic of the regulation of TAM polarization by functionally different proteins/extracellular vesicles, components of the gut microbiota/natural bioactive compounds, and ncRNAs (lncRNAs and miRNAs) demonstrated in CRC.

## 5. Clinical Role of TAMs in Colorectal Cancer

The prognostic and predictive role of TAMs in CRC has been the subject of intensive study in recent years, but the results are inconsistent [7,76,77,78,79,80,81,157]. The high heterogeneity of TAMs in CRC tissues and the diverse impact of phenotypically different cells on prognosis in this type of cancer are emphasized. The combination of different immune cell markers in TME (including TAMs), their evaluation in different areas of the tumor (stroma, invasive front, intraepithelial area), and at different stages of CRC development (e.g., intramucosal neoplasia, submucosal invasive CRC, and advanced cancer) has been studied [5,77,79,158,159,160].

### 5.1. In Vitro Studies

Ong et al. showed inhibition of tumor cell proliferation by TAMs, as well as production of both cytokines (IL-6, IFN-γ) and chemokines (IL-8/CXCL1/10, CCL2) that attract T cells and promote type 1 T cell responses [75]. In a study by Coletta et al., mononuclear monocytes cultured together with tumor cells or decellularized tumor matrix differentiated into a pro-tumoral macrophage phenotype, characterized by decreased expression of MHC-II and CD86, increased expression of CD206, and abundant release of pro-tumoral cytokines (e.g., IL-6, IL-10, TGF-β) and chemokines (e.g., CCL17, CCL18, and CCL22). As a result, macrophages were unable to present antigens to CD4 T cells. These data suggest that TME contributes to the defining pro-tumor profile of macrophages infiltrating CRC tissue with an impaired ability to activate effector T cell function [161].

### 5.2. Animal Models

The involvement of TAMs in the initiation and progression of CRC includes phenomena seen in other cancers, although, in the case of this cancer, the interaction between CRC cells and GI tract microbiota is emphasized [7,97,144,162]. These interactions involve, among others, the use of bacterial LPS as a trigger for the accumulation of monocyte-like macrophages in a chemokine-dependent manner and the generation of a precancerous inflammatory environment facilitating the development of cancer [144]. A significant increase in the production of pro-inflammatory cytokines (e.g., IL-1β, TNF-α, and IL-6) by macrophages was observed in the dextran sodium sulfate (DSS) colitis mouse model. These cytokines promoted the stemness of doublecortin-like kinase 1 (Dclk1)+ tufted cells, which may serve as a cellular source of cancer. Furthermore, macrophages with the CD11b+F4/80+Ly6Chigh phenotype have been shown to be key mediators of CAC initiation in mice [58]. In turn, studies by Li et al., also on CAC, showed that *Bifidobacterium breve* (*B. breve*) alleviated symptoms in various colitis models, delayed colonic neoplasia, and promoted phenotypic differentiation of immature inflammatory macrophages into mature homeostatic macrophages. The inhibitory effect of *B. breve* on cancer development would be to alter tryptophan metabolism by directing the differentiation of immature colonic macrophages [97].

### 5.3. In Vivo Studies

Variations in M1/M2 TAM ratios and their differing correlations with clinical data, including patient survival prognosis, have been described in CRC tissues [73,74,75,80,81,96,159,163,164]. Ong et al. demonstrated the predominance of pro-inflammatory TAM infiltration of CRC tissues, resulting in anti-tumor effects. These findings would explain the observations that high macrophage infiltration in CRC tumors correlates with a good prognosis for CRC patients [75]. Another study showed increased infiltration of both NOS2+ M1 and CD163+ M2 macrophages in the invasive tumor front (TF). Both types of macrophages were shown to be inversely correlated with the tumor stage. No differences in cancer-specific survival (CSS) were observed in CRC groups with different M1/M2 macrophage ratios. However, studies suggest that the presence of abundant M1 macrophages promotes survival in CRC patients, despite the presence of M2 macrophages [74].

Gulubova et al. show the prognostic significance of a low level of CD68+ macrophage infiltration as an unfavorable survival marker in CRC patients. They also showed a correlation between reduced CD68+ macrophage infiltration in tumor stroma with TGF-β expression [163]. A positive correlation between a high density of CD68+ TAMs (M1) and better OS was shown by Koezler et al. In addition, interactions between stromal CD68 and invasive tumor cells were frequent and mitigated the prognostically detrimental effect of tumor budding phenotype. Interestingly, in tumors with strong CD163 infiltration (M2), CD47 expression had a negative effect on survival [165]. This was supported by the studies of Li et al., where strong CD68+ expression in the tumor stroma correlated with longer overall survival (OS) in CRC patients. We should add that, among all the infiltrating immune cells examined, CD68+ macrophages dominated, and their number correlated with the number of other immune cells (e.g., stem cells, T and B lymphocytes), a lower expression of EMT markers, and a lower number of tumor buds. TAMs present at the invasive front of the tumor may, therefore, play a key role in counteracting the adverse effects of tumor buds on the course of CRC [158]. Also, in the study by Strasser et al., macrophages dominated among all immune cells. A lower number of CD206+ macrophages (M2) was demonstrated in CRC compared to distant normal colon mucosa, and an association between reduced CD206 expression and poor prognosis in CRC patients [77]. Studies by Pinto et al. showed a correlation between higher CD68 levels and lower CD80/CD163 ratio and reduced OS, but only in stage III tumors. Despite the low infiltration of pro-inflammatory CD80+ macrophages in CRC, a protective role of these cells has been demonstrated in relation to the risk of disease relapse [166]. Kou et al. demonstrated a prognostic and predictive role of both subpopulations of macrophages, i.e., CD86+ (M1) and CD68+CD163+ (M2). High expression of M1 macrophages (CD86+ and CD68+CD86+) and low expression of M2 macrophages (CD163+ and CD68+CD163+) were good prognostic factors for OS [80]. Väyrynen et al. showed that only a high M1/M2 density ratio in the tumor stroma was associated with better CSS, while the total density of macrophages in the intraepithelial or stromal areas of the tumor was not prognostic [159]. There are also studies suggesting a positive correlation between high levels of M2 macrophages (CD206+) and relapse-free survival (RFS) in CRC. Moreover, M2 macrophages are thought to play a role in limiting the metastasis process by influencing the maturation and normalization of blood vessels [81]. The study by Majid et al. suggests a significant prognostic role of high TAM density in stages I-III CRC (favorable 5-year RFS). However, a good prognosis was influenced by a simultaneous high density of T-cells in the tumor, whereas a poor prognosis was observed with a high number of TAMs and a low number of T-cells [167].

There are also studies that have shown inverse correlations between the intensity and/or location of TAMs within the tumor and prognosis in CRC, often depending on the stage of the tumor [73,96,164,168,169]. The majority of these studies focus on macrophages of the M2 phenotype. In one study, although a positive correlation was observed between a high number of peritumoral CLEVER-1/Stabilin-1+ (M2) macrophages and survival, patients with a high number of peritumoral or intratumoral M2 macrophages had a shorter disease-specific survival in stage IV CRC [168]. The presence of intense M2 macrophage infiltration (CD68+ or CD163+) in the tumor stroma was a poor prognostic factor and correlated with shorter disease-free survival (DFS) and OS, although the opposite tendency was observed at TF. An association between CD68+ TAMs and the relative risk of tumor recurrence was also observed, while the odds of cancer-related death were almost doubled in CD68+/iNOS- patients [164]. Ye et al. also showed that patients with fewer TAMs and regulatory T (Treg) infiltrating the stroma had longer DFS and OS, regardless of chemotherapy [73]. Another study demonstrated an association between CD163+ TAM (M2) infiltration at the TF tumor with EMT, the circulating tumor cells (CTCs) ratio, and poor prognosis in CRC patients [170]. The same research group observed a correlation between increased CD163+/CD68+ at TF (but not at TC) and shorter RFS and OS of CRC patients. Moreover, CD163+/CD68+ at TF was identified as a better prognostic factor compared with CD68+ at TF and/or CD163+. Polarized TAM2 secreted TGF-β to facilitate EMT, growth, proliferation, and invasion of CRC cells in in vivo and in vitro experiments [171]. Similarly, elevated serum levels and tissue expression of mannose receptor (MR) and CD163 (M2) were found to be significantly associated with shorter OS and were identified as an unfavorable prognostic factor. Furthermore, preoperative serum levels of MR and CD163 were found to correlate with those of carcinoembryonic antigen (CEA), carbohydrate antigen 19-9 (CA19-9), and cancer antigen 72-4 (CA72-4). The serum levels and tissue expression increased with lymph node metastasis (LNM) [169]. In the studies by Inagaki et al., an increase in the number of M2 macrophages, present mainly at TF, was observed in conjunction with tumor progression. High numbers of M2 and low numbers of M1 macrophages were correlated with LNM. The study concluded that the assessment of the M2/M1 ratio at TF offers a higher predictive value for LNM when compared with the study of single TAM phenotypes. However, the prognostic value of TAMs for the survival of CRC patients was not assessed in this study [79]. Recent studies have confirmed that patients with low expression of three M1 markers (NOS2/CXCL10/CD11c) and higher expression of three M2 markers (CD163/CD206/CD115) had a lower OS rate. The combination of markers was found to be a better prognostic factor than a single macrophage marker [160]. A recent pilot study in CRLM showed differences in the density of differently polarized macrophages (M1 and M2) depending on the tumor region. M2 macrophages were the dominant subtype both in the center and at the periphery of the tumor, with relatively higher expression at the periphery. The presence of the above-median CD68+myeloid-related protein 8-14 (MRP8-14)+CD86- (M1) expression was associated with poor OS (median 2.3 years) compared with below-median M1 macrophages at TC (median 6.41 years) [96].

The first meta-analysis on the prognostic role of TAMs showed that CD68 (TAMs1) expression is associated with poorer prognosis in numerous solid tumors, except for CRC [172]. A more recent analysis has shown that CRC with increased CD68 expression had a better prognosis for OS compared to other tumors (hazard ratio = 0.56 vs. 1.34). The proposed M2 macrophage markers were associated with poorer survival in epithelial tumors and melanoma [78].

A summary of the latest studies on the prognostic role of TAMs in patients with CRC is presented in Table 4.

Table 5 presents a set of the most common classically activated (M1) and alternatively activated (M2) markers detected in normal colon/rectum and present in CRC (TAMs), along with the definition of their biological effects in physiology and cancer.

## 6. Therapies Targeting TAMs in CRC

Therapies targeting TAMs in solid tumors (including CRC) aim to limit monocyte recruitment, deplete TAMs, induce cell reprogramming (M2 to M1), promote macrophage phagocytic activity, and/or disrupt the balance between macrophage recruitment and their effector functions. More promising are strategies that combine TAM-based therapies with conventional therapies [62,88,157,174,175,176,177]. TAM-based therapies are challenging because these cell populations are highly plastic and often adapt or become resistant to this type of treatment [62,174,175,176,177,178]. Myeloid-derived suppressor cells (MDSCs) and TAMs derived from them may be directly related to resistance to chemotherapy. It has been shown that in CRC mice treated with OXA, there was a significant reduction in the number of M-MDSC and M1 TAMs differentiated from MDSC. Administration of OXA, only in combination with TLR agonists, i.e., R848, reversed the functional orientation of MDSC toward M1 macrophages and enhanced the anti-tumor effect of the drug. Thus, a new immunological mechanism of OXA resistance was discovered, and the potential of the TLR7/8 agonist as a new immunological adjuvant in the chemotherapy of CRC patients resistant to this drug was demonstrated [178].

The most commonly mentioned signaling pathways that should be blocked to achieve an anti-tumor effect in CRC through repolarization of M2 macrophages to M1 include STATs (mainly STAT3 and STAT6) [94,103,104,179], NF-κB/MAPK [94,108], peroxisome proliferator-activated receptors (PPARs) [94], and HIFs [94,107]. In terms of blocking tumor-promoting signals and repolarization of TAMs, inhibition of the CSF-1R has been shown to preferentially eliminate macrophages with an inflammatory signature but spare macrophage populations expressing proangiogenic/cancer genes in mice and humans [82]. In turn, Lee et al. demonstrated in a mouse colon cancer model that an oral CSF-1R inhibitor called BPR1K871 specifically inhibited M2 macrophage survival with minimal effect on M1 macrophage growth. In vivo, oral administration of BPR1R024 mesylate (12 MsOH) delayed colon tumor growth in mice and reversed the immunosuppressive TME with an increased M1/M2 ratio [180]. Another oral CSF-1R inhibitor was a urea derivative called compound 21. It also demonstrated potent antiproliferative activity and inhibition of CRC progression by reducing macrophage migration, reprogramming M2 macrophages to the M1 phenotype, and enhancing anti-tumor immunity. Compound 21, as a single agent, showed significantly higher anti-tumor efficacy in vivo than PLX3397 [181]. Fang et al. also used magnetic hyperthermia in combination with a CSF-1R inhibitor called TAT-BLZ945 in liposomal form (termed TAT-BLZmlips) to repolarize macrophages in the CRC microenvironment to alleviate immunosuppression, normalize tumor blood vessels, and promote T-cell infiltration. Anti-tumor CD8+ effector T cells were increased after such treatment [182]. Therapeutic approaches to alter macrophage polarization not only target CSF-1R but also treat M2 macrophages with non-PEGylated and PEGylated liposomes containing IFN-γ. IFN-γ-containing liposomes were shown to be more effective than free IFN-γ in reducing arginase and NO levels, i.e., in altering M2 to M1 macrophage polarization in the mouse model [183].

A different therapeutic approach is immunotherapy based on CD47/signal regulatory protein α (SIRPα) signaling [175] or anti-CD47 [82,83,184]. In in vitro and in vivo models, the CD47 blocker ALX148 has been shown to stimulate anti-tumor properties of innate immune cells by promoting DC activation, macrophage phagocytosis, and shifting TAMs toward an inflammatory phenotype. ALX148 was generated by fusing a modified SIRPα D1 domain to an inactive human IgG1 Fc. ALX148 binds CD47 from multiple species with high affinity, inhibits wild-type SIRPα binding, and increases macrophage phagocytosis of tumor cells [184].

CD47/SIRPα blockade is expected to enhance the phagocytic activity of TAMs against cancer cells and exert effective anti-tumor activity. Clinical trials (phase 1/2) of this therapeutic approach in CRC are currently underway with the drug Hu5F9-G4 as a single agent in combination with cetuximab (CET) (anti-EGFR IgG1 monoclonal Ab, used in disseminated CRC) [175]. Humanized pan-allelic anti-SIRPα antibodies, called AB21 (hAB21), have also been developed, which bind human SIRPα protein with high affinity and block the interaction with CD47. These have been used alone or in combination with inhibitors of programmed cell death ligand 1 (PD-L1). The use of this therapeutic approach also increased the M1 population of macrophages. CD47/SIRPα blockade has been confirmed as a promising therapeutic intervention in the treatment of human malignancies, including CRC [185]. Two new differentiated antibodies, i.e., SIRP-1 and SIRP-2, which act in this axis by different mechanisms, are also promising, as demonstrated in in vitro models. SIRP-1 Ab directly blocks CD47/SIRPα, and SIRP-2 Ab acts by disrupting higher-order SIRPα structures on macrophages. Both Abs also increase phagocytosis in combination with tumor-opsonizing Abs, including a highly differentiated anti-CD47 Ab (AO-176) currently evaluated in phase I clinical trials [186]. Recent studies by Matusiak et al. suggest that the targets of anti-CD47 and anti-PD-L1 immunotherapy in CRC are IL-4I1+ macrophages, which may affect the phagocytic potential of TAMs. They also point to IL-4I1 as a potential new predictive marker for blockade of the programmed cell death protein 1 (PD1)/PD-L1 axis in this tumor [83].

The summary and characteristics of selected National Institutes of Health clinical trials on TAM-targeted therapy in the treatment of various types of cancers (including CRC) are presented in numerous reviews [62,88,176,177,187]. In the strategy targeting TAM recruitment in metastatic CRC therapy, some chemokine inhibitors are considered as targets, e.g., CXCR4-CXCL12 inhibitors alone or with durvalumab (anti-PD-L1 Ab) (phase I) [176]; C-C motif chemokine receptor 2/5 (CCR2/5) (agent BMS813160) with nivolumab (anti-PD1 Ab); or paclitaxel (phase I/II) [62] and pembrolizumab (CCR5 antagonist) (phase I completed) [187,188].

Among immunotherapies targeting TAM repolarization in CRC, phase I trials have been completed for pexidartinib (CSF-1R antagonist) with durvalumab and CCR5 inhibitors as monotherapy, as well as phase I trials and recruitment of CT-0508 (chimeric antigen receptor macrophages) as monotherapy (reviewed in [177]). Another set of TAMs as promising targets for CRC treatment was presented by other authors (reviewed in [88]). These are TAM-reprogramming drugs, e.g., anti-MARCO IgG [189], tasquinimod (S100A9 inhibitor) [190], ribonuclease T2 (RNASET2) [112], or enhancer of zeste 2 polycomb repressive complex 2 subunit (EZH2) inhibitors (EPZ6438 and GSK126) [191].

Drugs that inhibit TAM survival and depletion include pexidartinib (PLX3397) (CSF-1R inhibitor) [192], trifluridine/tipiracil (FTD/TPI) [193], OXA [193], M2pep [194], RG7155 (anti-CSF-1R Ab) [195], and lenvatinib (tyrosine kinase inhibitor) [196].

Drugs that inhibit TAM recruitment include the following: NT157 (IGF-1R inhibitor) [179] and AMG820 (Amgen) (anti-CSF-1R Ab) alone [176,197] or in combination with pembrolizumab (phase I/II completed) [198], and pexidartinib alone or with durvalumab [176]. Some studies suggest that pexidartinib alone inhibits immunosuppression by reducing the percentage of the TAM population, but not their polarization. In contrast, triple combination therapy (pexidartinib, oncolytic adenoviruses, and anti-PD1 Ab) in mice inhibited TAM recruitment to the tumor and reprogrammed TAMs toward the M1 phenotype, in which the percentage of CD206+ TAMs was reduced [199].

The latest review presents TAM-based drugs used in CRC immunotherapy as follows: CET [200], Maraviroc (CCR5 inhibitor) [201], or regorafenib (REG) (tyrosine kinase inhibitor) [202] (reviewed in [157]). CET repolarized TAMs from an M2-like to an M1-like phenotype, primarily by inhibiting IL-6 expression via the NF-κB and STAT3 pathways [200]. Maraviroc effectively polarized “normal” macrophages into tumor-killing macrophages that produced ROS and IFN-α [201]. In a phase I clinical trial testing maraviroc as monotherapy in advanced CRC, selective tumor cell necrosis was observed in CRLM [203]. REG treatment significantly reduced the infiltration of immunosuppressive macrophages and Tregs into the TME, while anti-PD1 Ab significantly increased intratumoral IFN-γ levels. The drugs acted synergistically, inducing permanent M1 polarization and a reduction in Tregs, which may explain sustained tumor suppression [202].

Table 6 summarizes both established and novel candidates that show promise in CRC treatment strategies based on the action of TAMs.

## 7. SRIF System as Regulator of the Macrophages

Numerous NE peptides (e.g., cholecystokinin, bombesin, SST), and growth factors (e.g., PDGF, EGF, IGF-1, FGF-2, VEGF, TGF-β), which are integral components of well-known signaling pathways, play an important role in the pathogenesis of CRC [6,11,12,13,14]. Correlations have been observed between the expression of neuropeptides from the GI tract and changes in immune cells during inflammatory processes. Furthermore, the interaction between neuropeptides/amines in the GI tract and gut microbiota is a critical factor in the pathogenesis of inflammatory bowel diseases (IBD) [14].

Several studies indicate that macrophages of the GI tract, including the normal colon and rectum, produce SST, which is one of the immunoregulatory cytokines in inflammation [16,17,18,19,20]. SST2 is the main receptor type for the autocrine and paracrine effects of SST in immune cells, including macrophages [19,21,22,23]. While the immunosuppressive effects of SST are well-documented, the exact role of SRIF system components expressed by TAMs in CRC is poorly understood.

The SRIF system comprises seven genes encoding two peptide precursors, SST (or SRIF) and CST, as well as five receptors (SSTRs) [15]. SST, first described in rat [204] and sheep hypothalamus extracts [205], is a well-known neuropeptide widely distributed in the central nervous system (CNS) and peripheral nervous system (PNS); it regulates the endocrine system and affects neurotransmission through interaction with five SSTRs.

SST is formed from a 92-amino acid preprosomatostatin precursor, which is processed and utilized at the COOH-terminal segment to produce two biologically active forms, SST-14 and SST-28. In the hypothalamus and other parts of the CNS, the PNS, and pancreatic δ cells, SST-14 is mainly expressed, while SST-28 is mainly represented by the remaining D-cells in the GI tract (reviewed in [206]). In vivo and in vitro studies confirm the expression of SST/SSTRs in the lymphatic system and the interactions between these neuropeptides and their role in immune regulation [207,208,209,210,211,212].

### 7.1. Somatostatin (SST)

In the GI tract, the majority of SST production occurs in the mucosa (>90%), mainly in the stomach, duodenum, and jejunum, with <10% in the submucosal and muscle layers. In the mucosa, SST is localized mainly in epithelial enteroendocrine cells (EECs) [213,214]. In the GI tract, it acts as a pan-inhibitory peptide in the processes of endocrine and exocrine secretion. It also affects motility, blood flow, and intestinal absorption. SST has anti-inflammatory effects and a direct immunomodulatory effect [17,215,216,217]. It inhibits many immune functions, including lymphocyte proliferation, immunoglobulin production, and the release of pro-inflammatory cytokines, such as IFN-γ (reviewed in [17]). It is considered an important regulator of the innate immune system during *Helicobacter pylori* (*H. pylori*) gastritis. The attenuation of inflammation by IL-4 is due to the mechanism by which this cytokine stimulates the release of SST from gastric D cells [210,218]. The immunoregulatory properties of SST were also demonstrated by its effect on DCs in gastritis. SST and its even more potent analog, octreotide (OCT), inhibited IL-12 release by *H. pylori*-stimulated DCs. In addition, the production of SST in DCs and the increased secretion of SST by IL-4 were demonstrated. These studies suggest that the immune regulation of DCs by SST may also affect non-gastric tissues and be involved in the anti-inflammatory effects of T helper cell type 2 (Th2) cytokines. In vivo studies confirmed the colocalization of the DCs marker (CD11c) and SST in the lamina propria of mouse stomachs infected with *H. pylori* [219].

The anti-inflammatory effect of SST is partially mediated by monocyte/macrophage deactivation [220] and the inhibition of IFN-γ release by T cells [215,221]. The role of SST in modulating macrophage function in carcinogenesis is emphasized. It can increase macrophage cytotoxicity against cancer cells, but also block macrophage neoplastic activity induced by recombinant IFN-γ [222].

The production of SST, among other neuropeptides, has been confirmed in numerous human lymphatic organs (e.g., lymph nodes, tonsils, appendix, spleen, and thymus), located in different compartments and in different cell types. The source of SST in these organs was also nerve endings [209,211,223].

Much attention has been paid to studies on the cellular localization and role of neuropeptides (including SST) in the thymus of many species [212,223,224,225,226,227]. Immunoreactive-SST-28 was localized in the thymus of neonatal and adult rats, appearing as epithelial or NE-like cells in the thymic corticomedullary junction [224]. In the mouse thymus, high expression of SST was found in both cortical and medullary epithelial cells, and SST2 in thymocytes [225]. In birds, SST and its receptors have been shown to be expressed in both thymic epithelial cells (TECs) and thymocytes, suggesting intratissue neuroendocrine interactions [227]. Furthermore, in the avian thymus, SST expression appears to change with tissue age [226].

Intracellular pathways mediating SST-dependent activities in the thymus have been implicated in the maturation and selection of the T-cell repertoire [223,228]. Expression of SST mRNA as well as SSTRs (SST1, SST2A, and SST3) has also been demonstrated in normal human thymic tissues and in isolated TECs [229]. No SST mRNA expression was detected in human thymocytes [228], but CST expression was demonstrated in these cells [230]. This suggests that SST produced by a subset of secretory TECs [229] and endogenous CST produced by thymocytes [230] may affect thymic cell populations in a paracrine and/or autocrine manner. Thus, SSTs and SST-like peptides may be involved in the regulation and selection of T cells in the thymus [228]. It has also been proven via a mouse in vitro model that SST increased the number of fetal thymocytes and their maturation in the fetal thymus. In addition, SST induced thymocyte migration. It also increased the cellular proliferation of all splenocytes but inhibited the proliferation of mature (adult) thymocytes and purified splenic T cells. The role of SST in immune regulation and thymic development in mice has been investigated [225].

SSTs and SSTRs have also been implicated in the involution of the human thymus. An inverse correlation between the number of SST2As and thymic age has been demonstrated [223]. Other studies suggest that exogenous SST-14, although not involved in the control of physiological thymic involution, inhibits total body growth and induces thymic mass loss by reducing true lymphoid tissue. SST-14 alters the relationship between thymocyte subsets, modulating their development and maturation [231]. Thus, SST expression occurs in the thymus of mammals (humans, rodents) and non-mammals (birds, amphibians, and teleosts), where it plays an important role in the regulation of the immune system [211,226].

#### Somatostatin (SST) Expression in Macrophages

In normal rat thymus, co-expression of SST with markers of antigen-presenting cells (APCs) (e.g., MHC-II molecule) was observed on at least one-third of the APC membrane. SST-positive cells were diffusely distributed in all three zones of the gland, i.e., cortical, medullary, and corticomedullary (with most in the corticomedullary zone). Morphologically, SST-positive cells were round, oval, or elongated with an irregular angular contour. SST in the thymus is thought to be involved in the action of APCs, particularly in the expression of MHC-II and all coreceptor molecules and cytokines that are important for T cell activation, positive selection of T cell clones, and induction of apoptosis of defective T cells (negative selection) [212].

In a mouse model, secretion of the precursor and mature SST-like molecule-14 has been demonstrated by activated macrophages from granulomas and in splenocytes isolated from mice infected with *Schistosoma mansoni* (*S. mansoni*). Macrophage-like cells from the P388D1 and J774A.1 cell lines also expressed SST precursor mRNA. These data suggest that SST derived from macrophage-like cells is an inducible component of the innate immune system that inhibits IFN-γ production by T cells present in the granuloma [207,208,215]. Like T and B cells, granuloma macrophages also expressed SST-14 mRNA in response to LPS, IFN-γ, IL-10, and several other inflammatory mediators [16,215,221]. SST production by both activated and nonactivated peritoneal macrophages has also been confirmed [232]. SST and its receptors are also present in rat liver macrophages (Kupffer cells, KCs) [233] and in human macrophages [23].

Somatostatin has direct anti-inflammatory effects and also participates in the anti-inflammatory effects of glucocorticoids. As shown in the macrophage cell line RAW 264.7, SST promoted a time- and dose-dependent increase in [3H]dexamethasone binding to macrophages. Exposure of cells to 10 nM SST for 18 h promoted a 2-fold increase in the number of glucocorticoid receptor (GR) sites per cell without significant change in affinity. This occurred through a mechanism involving increased glucocorticoid binding and signaling, achieved by reducing calpain-specific cleavage of heat shock protein 90 (Hsp 90) associated with GRs [234].

### 7.2. Somatostatin Receptors (SSTRs, SST1-5)

SST receptors are expressed at varying densities in NE cells, as well as in many non-neuronal and non-endocrine cells [235]. Among the SSTRs associated with the function of the lymphatic system, particularly with monocytes/macrophages in health and disease, the role of SST2 (especially its isoform SST2A) is most often emphasized [16,19,21,23,215,221,228,230,236,237]. The *sst2* gene can be alternatively spliced to produce two receptor proteins, SST2A and SST2B, which differ in length and sequence of their carboxyl termini. Human tissues almost exclusively contain the unspliced SST2A variant, whereas both spliced forms have been identified in rodents [238]. The SST2B variant encodes a protein that is 23 residues shorter than predicted from the SST2 sequence and differs by 15 amino acids at the C-terminus. It binds SST with high affinity when expressed in mammalian cells [239].

There is a high degree of sequence identity in the phosphorylated regions of SST2A (rat and human SST2A receptors differ by only two residues) [238,240]. Regulation of the SST2A subtype, which is most abundantly expressed in human neuroendocrine tumors (NETs), is the best-known receptor subtype and the only one that has been studied in patients [238,241]. Analysis of the SST2A phosphorylation status is important because it is a sensitive indicator of the efficacy of SSAs in the diagnosis and therapy of patients with NETs [241]. The generated rabbit polyclonal antibody (R2-88) against a unique sequence in the COOH-terminal region (aa 339–359) enabled the visualization of a common region for human, rat, and mouse SST2A [236,242]. Reubi et al. were the first to demonstrate the immunohistochemical (IHC) localization of SST2A in the lymphatic and nervous components of the GI tract [236]. And Gugger et al. provided the first description of the IHC localization of the SST2A protein in EECs present in the epithelium of the non-neoplastic human intestine (mainly in the midgut). Moreover, SST2A+ cells were found throughout the GI tract in the myenteric and submucosal plexus [243].

In the early 1990s, the presence of specific SSTRs on circulating human mononuclear cells and erythrocytes was demonstrated [244]. The ability of SST to bind to both the monocyte fraction and lymphocyte fractions obtained from the peripheral blood of healthy volunteers was confirmed [245]. Tissue expression of SSTRs has been observed in normal lymphatic organs, where it occupies different compartments [236,242,246].

In human gut-associated lymphatic tissue of the colon, SSTR expression was found mainly in secondary follicles. In the human spleen, SSTRs were found in the red pulp and in the thymus, mainly in the medulla [24,242,246,247]. Most SSTRs in lymphoid organs were most likely of the SST2 subtype [242,246]. A detailed study of SST2A in the human colon through receptor autoradiography with 125I-[Tyr3]-OCT and IHC confirmed its presence in neural elements of the myenteric plexus, submucosal plexus, and germinal centers of lymphoid follicles. Strong SST2A expression was also detected in the mucosa, while its absence was found in the smooth muscle of the GI tract [236]. This confirms previous studies using in situ hybridization in the rat GI tract, but SST2A mRNA expression in lymphoid follicles in the small intestine was more present in the corona than in the germinal centers. In the colon, the strongest mRNA expression of all SSTRs was observed compared to the rest of the GI tract [248].

The expression of three SSTRs (SST1, SST2A, and SST3) was demonstrated in the normal human thymus. Cultured thymic TECs selectively expressed SST1 and SST2A and responded to SST and OCT administration by inhibiting cell proliferation. This indicated a paracrine/autocrine role of SST and its receptors in the regulation of thymic growth and the microenvironment [229]. Moreover, the expression of several SSTRs was detected in human thymocytes during thymic development. They were activated after binding to SST produced by TECs. Thymocytes selectively produced SST1 and SST2A. They regulated the proliferation and maturation of immature thymocytes, including their migration through the thymic stroma, cytokine production, and export from the thymus [223,228]. SST2 expression has also been demonstrated in mouse thymocytes [225].

Quantitative real-time PCR (RT-PCR) showed significantly higher SST2A mRNA expression in the spleen, while SST and SST3 expression were dominant in the thymus. The highest SST2A density in the spleen is consistent with the in vivo uptake of [111In-DTPA-D-Phe1]-OCT, which is considered to be an SST2-preferring ligand. IHC confirmed the preferential expression of SST2A on the cells of the microenvironment, and SST3 on the lymphoid cells of these organs [24].

#### Somatostatin Receptor (SSTR) Expression in Macrophages

Systemic examination of various normal human tissues and organs confirmed marked immunoreactivity for SST2A in monocytes and macrophages [21]. The presence of mRNA for SST2 was demonstrated in various granuloma cells and on macrophage-like cell lines. This receptor was hypothesized to mediate the action of SST on T cells, on which two subtypes of SST2 (SST2A and SST2B) were present. The SST2A isoform was considered the dominant receptor mediating the effects of SST on the immune system [16,215,221]. In murine schistosomiasis, SST modulated the secretion of IFN-γ from T cells, a process that also required the presence of SST2. It is now known that the predominant isoform of SST2 on inflammatory cells is SST2A (99%) [21,22,215,221].

In purified mouse peritoneal macrophages using radioisotope-labeled SST-14, two functional subtypes of SSTRs, i.e., SST1 and SST2, were identified and characterized. Furthermore, RT-PCR analysis revealed the presence of mRNA for all five SSTRs in these cells. The involvement of SST1 and SST2 in the inhibition of adenylyl cyclase (AC) was confirmed [220]. Recent studies have confirmed the expression of SST2 in different cell types (including lung macrophages) in various organs of mice, and at different developmental periods [249].

The presence of SST and its receptors was also confirmed in rat KCs [233] and human macrophages [23,237]. Dalm et al. confirmed the selective expression of SST2 mRNA in human monocytes, macrophages, and DCs. However, these cells did not produce SST, but CST mRNA. LPS stimulation of these cells increased the expression of only SST2 and CST mRNA. Furthermore, a significant increase in SST2 and CST mRNA was observed during the differentiation of monocytes into macrophages and DCs. Therefore, the selective and induced expression of CST and SST2 was demonstrated in human macrophages derived from monocytes, suggesting a role for the CST-SST2 system rather than the SST-SST2 system in these types of immune cells of the human immune system [230]. Armani et al. demonstrated that human macrophages differentiated from monocytes derived from PBMCs express SST1 mRNA in addition to SST2 mRNA. Both transcripts were localized on cell membranes and had active binding sites. Functionally, activation of these SSTRs by multiligand analogs SOM230 and KE108, but not by SST-14 or CST-14, reduced macrophage viability. These results suggest that the SRIF system, through both mentioned SSTRs, exerts immunosuppressive effects in human macrophages. This finding may have therapeutic implications [23].

Other studies have shown the presence of both endogenous SST and CST in rat liver KCs. Both peptides would act in an autocrine manner, regulating the distribution of SSTRs. Quiescent KCs were found to express SST-14 mRNA, whereas IHC studies have confirmed the presence of only SST3 and SST4. SST1 and SST2A were detected using western blotting. Stimulation of cells with LPS has been shown to activate the expression of SST2, SST3, and SST4 [233].

The most abundantly represented SSTR subtype on macrophages of various organs in health and disease is considered to be SST2A [19]. Selective expression of SST2A mRNA has also been demonstrated in human thymic CD14+ cells belonging to the monocyte-macrophage lineage. The heterogeneous expression of SSTRs in the human thymus on specific cell subsets, along with the endogenous production of SST and SST-like peptides, emphasizes their role in mutual interactions between the main cellular components of the thymus involved in T cell maturation [228].

### 7.3. SST/SSA Actions in the Monocytes/Macrophages

SST is considered one of the immunoregulatory cytokines in inflammatory conditions produced by macrophages, as shown by animal models (mice, rats) and in vitro studies [16,18,19,22,215]. It modulates some immune functions that are not classified as secretory or proliferative. These functions include the inhibition of human NK cell activity and human neutrophil chemotaxis, the reduction of phagocytic activity of human monocytes and macrophages, and T cell migration [18,19,22]. SST is a component of the peptidergic neurotransmitter complex in the thymus, produced by thymocytes and microenvironment cells (except nerve fibers). It has been demonstrated that SST, produced by secretory TECs, inhibits proliferation, differentiation, migration, and cytokine production by thymocytes [20]. The immunosuppressive effect of SST on macrophages has been suggested to occur through binding to SST2 and SST1 [19,23,220].

In a rat macrophage model, SST at concentrations of 10^−9^–10^−7^ has been shown to have a stimulating effect on antibody-dependent cellular cytotoxicity (ADCC) in rat peritoneal macrophages. This was due to the increased activity of the Fc receptor and the generation of active oxygen species. Moreover, SST at concentrations stimulating ADCC reduced intracellular cAMP production and gradually increased the level of cGMP. In contrast, higher concentrations of SST (10^−6^–10^−7^ M) inhibited ADCC in macrophages by activating both AC and Ca2+ influx [250].

SST has been shown to inhibit the secretion of numerous factors by immune cells (including macrophages) [18,251,252,253]. In physiological doses, SST (10^−10^–10^−7^ M) has been demonstrated to regulate the response of monocytes/macrophages isolated from human blood (healthy volunteers). Directly, this peptide inhibited the secretion of TNF-α, IL-1β, and IL-6 in LPS-activated cells as well as the chemotactic response of neutrophils to IL-8. The reduction of IL-8 production by monocytes was indirect through the effect of SST on the secretion of TNF-α and IL-1β. The suppression of monocyte activity was further confirmed by the reduction of the expression of the MHC class II cell surface receptor (HLA-DR) on their cell membranes [253]. SST derived from macrophages inhibited IFN-γ release by granuloma cells in *S. mansoni* infection [208]. It has been generally suggested that SST may be an important factor in controlling the granuloma inflammatory response in murine schistosomiasis [16,215,221].

In a rat model, SST and OCT have been shown to modulate the release of hydrogen peroxide (H_2_O_2_), NO, and TNF-α by peritoneal macrophages [251] and by KCs in cirrhotic livers, in a dose-dependent manner [252]. In mouse peritoneal macrophages, inhibition of AC by SST has been demonstrated, involving only SST1 and SST2 expressed on these cells. This is a pioneering study on the ability of SST to bind to macrophages via specific SSTRs. SST1 was characterized by high affinity and low capacity, and SST2 by low affinity and high capacity [220]. Another study showed that SST at doses of 10^−12^–10^−8^ M significantly inhibited the production of IL-6 and IFN-γ by peritoneal macrophages. Therefore, it was suggested that neuropeptides derived from macrophages and nerve fibers act as immunomodulators, mediating changes in the cytokine production pattern [232]. Additionally, SST produced by peritoneal macrophages participates in the apoptosis process by increasing the activity of p53, Bcl-2, Fas, and caspase-8, while decreasing the expression of iNOS and NO production [254]. The application of OCT in rat KCs resulted in a reduction of proapoptotic proteins, which was accompanied by an early increase and a late reduction of antiapoptotic proteins and a decrease in the expression of caspase-3 (LICE). The combination of LPS and OCT produced a similar effect, except for the late increase in LICE expression, probably caused by the late increase in bax and bcl-xS proteins [255]. The same research group also demonstrated a significant inhibitory effect of LPS and OCT (single or in combination) on the production of TGF-β1 by rat KCs [256].

SST administration at a high concentration (10^−6^ M) inhibited the migration of macrophages induced by *Leishmania major* promastigote [257]. The ability to inhibit the migration, phagocytosis, and survival of macrophages by SST is, therefore, emphasized [220,254,257]. In addition, SST and its analogs have been shown to modulate the production of inflammatory mediators produced in KCs. Studies by Valatas et al. demonstrated that OCT reduced IL-12 production and increased IL-13 production by unstimulated hepatic KCs. In LPS-activated KCs, this analog inhibited TNF-α production without modifying TNF-α mRNA expression and reduced iNOS expression and NO production. These effects were abolished after pretreatment with wortmannin, suggesting that OCT may act by interfering with PI3K pathways [258].

### 7.4. Macrophages and SRIF System in CRC—A Gap to Be Filled

One of the effects of the anti-tumor activity of the SRIF/SST system in CRC is the interaction between cancer cells and other TME cells (reviewed in [206,259]). Endogenous SST is also one of the regulatory neuropeptides affecting immune system cells [20,24,228,229]. Recent studies by Wu et al. confirmed the significant influence of the colonic immune system on the TME. Interestingly, they showed that moderate infiltration of stromal and immune cells is an important protective factor for patients with CRC, which was not found in other cancers. The SST gene belonged to the differential expression hub genes for both the moderate stromal cell infiltration group and the immune group. These genes exhibited a stronger correlation with tumor purity and infiltration of B cells, T CD8+ and CD4+ cells, macrophages, neutrophils, and DCs. Proteins encoded by these genes, including SST, have also been detected in CRC cells [260].

Although much is known about the role of TAMs in CRC, there is still a lack of studies on the role of SRIF system component expression in macrophages/TAMs present in the CRC microenvironment. However, there are studies showing the inhibitory effect of SST on the inflammatory process in the course of IBD, mainly ulcerative colitis (UC) and Crohn’s disease (CD) (reviewed in [216]). Patients with long-term IBD have a 2.4-fold higher risk of developing CRC than the general population, both in UC and CD [261]. Chronic inflammation causes oxidative stress-induced DNA damage, resulting in the activation of tumor-promoting genes and the inactivation of tumor-suppressor genes [262]. Long-term chronic inflammation is linked to an increased risk of CAC in UC patients. A detailed review also describes the role of macrophages in CAC. Increased macrophage infiltration results in the production of a large number of pro-inflammatory cytokines and the promotion of tumorigenesis. M1 polarization is effective in preventing tumor growth after CAC formation, while M2 polarization promotes tumor growth [263].

Animal models of IBD were used to investigate the role of the SRIF system in the control of the immune system in CAC. A decrease in intestinal macrophage and Th2 lymphocyte activity (decrease in IL-1β, IL-6, and IL-10 levels) was observed in the intestines induced by DSS in non-acute colitis (equivalent to non-acute UC), which was enhanced by SST-14 and OCT administration. However, neither SMS-14 nor OCT reduced mucosal inflammation or macroscopic disease activity [264]. A study on TNBS-colitis (equivalent to CD) as a model of Th1 disease showed increased production of TNF-α, IL-1β, and IFN-γ in colonic submucosal macrophages, and OCT treatment reduced all these effects [265]. In trinitrobenzene sulfonic acid (TNBS)-induced colitis in rats, an increase in SST-producing cells was observed in the colon tissues compared with controls, and a positive correlation between the density of these cells and the number of macrophages/monocytes and mast cells was observed [266]. In DSS-induced colitis in rats, a lower density of SST-positive cells was observed compared to the control group, similar to patients with IBD. The changes in all EECs were accompanied by an increase in the density of mucosal leukocytes, T and B lymphocytes, macrophages/monocytes, and mast cells as compared to the control group. Regardless of the IBD model, these studies confirm the cooperation between peptides produced by EECs (including SST) and immune cells (including macrophages) in IBD [267].

A study of CD68 expression (M1 macrophages) in glioblastoma multiforme (GBM) showed <10% to >50% microglia/macrophages in tumor samples, with the vast majority being resident microglial cells (equivalent to TRMs). A strong IHC reaction to SST2A was observed in the endothelial cells of proliferating vessels, in neurons and their processes, and only a weak one in isolated microglial and tumor cells. Although SST2A imaging showed areas of increased accumulation of the marker in each patient, this did not correlate with IHC results [27]. In another study, SST2A was also not detected in CD68+ macrophages bordering the necrotic area in glioblastoma [268].

### 7.5. SRIF System and TAM Interactions in CRC Diagnosis and Therapy

With the use of (In)111 pentetreotide (octreoscan), i.e., radioactively labeled SSA with high-affinity binding to SSTRs, it is possible to demonstrate many sites of SSTR expression. The detection of a large number of SSTRs in tumor macrophages, also in non-endocrine tumors, may be helpful in diagnosis, initial staging assessment, or prognosis of treatment effects. In granulomatous diseases, overexpression of SST2 binding sites has been detected in epithelioid and giant cells (as reviewed in [235]).

In the aforementioned glioblastoma multiforme, studies on the expression of SST2A and the TAM1 marker (CD68) showed only a low expression of SST2A on microglia or infiltrating macrophages. Hence, imaging and treatment strategies targeting SST2A do not seem promising in this type of cancer [27]. However, there was a case report of a patient with GBM (grade IV) who experienced local relapse after conventional therapy, and achieved a complete objective response with well-tolerated concomitant administration of SST/slow-release OCT, melatonin, retinoids dissolved in vitamin E, vitamin D3, vitamin C, dopamine 2 (D2) Receptor agonists, and Temozolomide [269].

Significant inhibition of the growth and progression of experimental liver metastases (including those from colonic adenocarcinoma) by OCT has been documented. However, the mechanisms of this effect were unknown [270]. In addition, increased activity of the liver reticuloendothelial system (now the phagocytic system) was demonstrated in OCT-treated rats compared with control animals 1 and 2 weeks after hepatectomy. OCT treatment significantly reduced the growth of both fibrosarcoma and colonic adenocarcinoma cells in the regenerating liver compared to the control group. The results of the study show that OCT inhibits both liver regeneration and tumor growth after partial hepatectomy [271]. This experimental evidence suggests that OCT may have an inhibitory effect on the development of liver metastases when used as an adjuvant to surgery in CRC [272]. In general, however, SSA therapy has been disappointing in the treatment of advanced malignancies. Improvement in the treatment of solid tumors may only come from the combination of SSAs with cytotoxic drugs or other hormones, both in advanced malignancies and as adjuvant therapy [273].

Guo et al. showed that a derivative of SST (smsDX) significantly attenuated TAM-stimulated proliferation, migration, and invasion of prostate cancer (PC). IHC examination revealed overexpression of NF-κB in both PC tissue samples and benign prostatic hyperplasia with chronic inflammation, and was positively correlated with macrophage infiltration. It was shown that the underlying mechanism of this phenomenon is NF-κB signaling, which plays an important role in macrophage infiltration. SmsDX inhibited the paracrine loop between TAMs and PC cells and may be a potential therapeutic agent for PC [26].

SSAs have been tried in patients with acute or chronic pancreatitis [274,275], but the clinical effects were not encouraging either. No benefit of OCT administration was found in terms of progression or treatment outcomes [274]. In pancreatic NETs, however, radiolabeled somatostatin analog/peptide receptor radionuclide therapy (PRRT) demonstrated the preservation of SST2A in the majority of patients (87%). In the PRRT group, the density of M2 macrophages (CD163+) was higher, and they tended to adopt epithelial morphology. Neoadjuvant PRRT in these patients was associated with reduced tumor diameter, increased percentage of stroma, preserved SST2A expression in most cases, and increased density of M2 macrophages [28].

Macrophage membranes can be used for detoxification and sequestration of pro-inflammatory cytokines due to their specific interaction with inflammatory factors and endotoxins [25]. Hence, macrophage membrane-coated nanoparticles have also been used to develop various therapeutic forms [276]. An example of the use of a macrophage membrane–polymer hybrid biomimetic nanosystem is its application in the treatment of chronic pancreatitis [25]. The system consists of delivering membrane-coated poly(lactic-co-glycolic acid) (PLGA) macrophages containing SST. It increases the bioavailability of SST, reduces the expression of chronic pancreatitis-related factors, and attenuates the severity of pancreatic histological changes. This enhancement of the therapeutic effects of SST demonstrated the superiority of integrating the synthetic polymer with biological membranes in the design of nanoplatforms for advanced peptide delivery. All the potential applications of bioinspired nanosystems in cancer diagnostics and therapy, and the latest advancements in this field, have been the subject of excellent reviews [276,277]. Separate reviews highlight the role of TAMs in anticancer nanomedicine [278].

Figure 2 shows a schematic summary of the available knowledge on the role of the SRIF system in modulating macrophages as representative immune cells in normal organs, certain inflammatory conditions, as well as cancers (CRC, PC).

## 8. Concluding Remarks and Future Directions

Currently, both the positive and negative roles of macrophages in health and disease are relatively well described. However, ongoing studies continue to investigate the role of these cells in cancer prognosis, including CRC, and the development of new forms of anti-tumor therapies. This review contributes to the current knowledge of the role of TAMs in CRC by providing a detailed description of emerging regulatory factors in this heterogeneous group of cells. It also discusses the clinical implications of assessing not only the phenotype but also the density and localization of TAMs in CRC tissues.

In comparison to normal mucosal macrophages, those present in CRC exhibit an altered phenotype. Many immunometabolic regulatory factors (e.g., cytokines, chemokines, LPS, SCFA, PCSK9, natural bioactive compounds), oncoproteins (e.g., HMGA2, MFHAS1), transcriptional/translational factors (e.g., CPEB3, TCF4), enzymes (e.g., PCSK9, CTSK, PKN2, RNASET2), enzyme agonists (MET), components of known signaling pathways (e.g., AMPK, EGFR, STAT3/6, mTOR, NF-κB, MAPK/ERK, and HIFs), and numerous epigenetic factors (ncRNAs) are involved in shifting the polarity of TAMs toward the M1 or M2 phenotype. Hence, significant heterogeneity characterizes TAMs in CRC tissues, with phenotypically different cells exerting a dualistic effect on prognosis. A high density of M1 macrophages within the tumor has been shown to be associated with improved survival outcomes for CRC patients, while a prevalence of M2 macrophages is linked to a less favorable prognosis. Consequently, for a more accurate prognostic assessment in CRC patients, the simultaneous testing of multiple different tissue markers of TAMs in the tumor is increasingly suggested. Understanding the role of various factors that regulate macrophage activity and polarization is essential to elucidate the molecular mechanisms that control cell–cell interactions within the CRC TME. Animal and in vitro models demonstrate that interactions of CRC cells with the GI tract microbiota (e.g., *F. nucleatum*, *E. coli*, *A. muciniphila*, and *B. breve*) play an important role in the initiation and progression of CRC.

There is a growing number of potential candidates for TAM-based therapeutics in CRC that act by different mechanisms. These strategies primarily aim to reprogram TAMs from an M2-like phenotype to an M1-like phenotype, inhibit M2 macrophage formation/activity, or promote M1 macrophage formation. These include known and novel biomolecules, numerous ncRNAs, and some natural bioactive compounds. While the debate on the efficacy of TAM-based treatment in CRC is still open, we believe that at least some of the newly described regulatory factors will be used as targets for anti-tumor therapy in CRC.

This review also collected data on the regulation of the immune system by the SRIF system, including the expression of some of its components (mainly SST and SST2) by macrophages, and the mechanisms of autocrine/paracrine action in physiological processes of the normal colon, in CAC, and CRC. Monocytes, macrophages, and TAMs are both sources and targets of the SRIF system in inflammation and cancer, including CRC. The effects of SST and SSA include the modulation of pro-inflammatory cytokine production (e.g., IL-1β, IL-6, TNF-α, IFN-γ) and the inhibition of apoptotic proteins in normal tissues and certain inflammatory conditions. In CRC, the SRIF system can both enhance macrophage cytotoxicity against cancer cells and directly inhibit tumor growth and liver metastasis, although the mechanisms are only partially understood. Notably, current research indicates that the direct inhibition of TAM-stimulated proliferation, migration, and invasion of cancer cells using SST analogs is unique [26].

Further research is needed to understand the mechanisms of action of SST on macrophages in the normal colon and rectum, as well as to investigate the role of this system in recruiting TAMs to the CRC TME, in reprogramming/polarizing macrophages, and in supporting the anti-tumor role of TAMs in CRC.

## Figures and Tables

**Figure 1 ijms-26-05336-f001:**
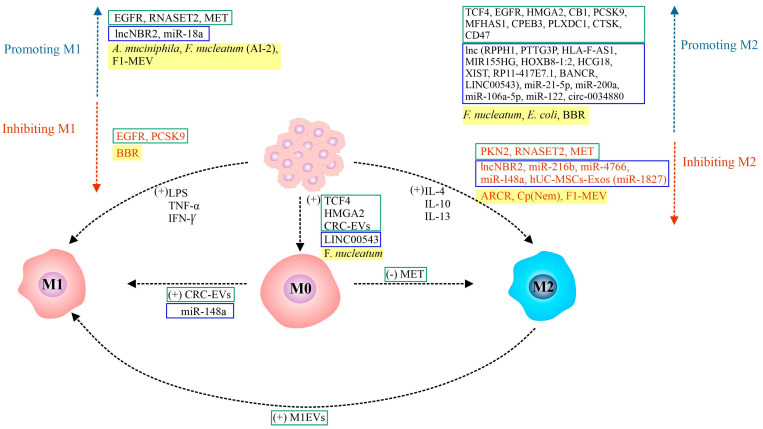
Signaling molecules regulating macrophage polarization in CRC: Undifferentiated macrophages (M0) can be polarized into two types: classically activated macrophages (M1) and alternatively activated macrophages (M2). The polarization of TAMs into M1 (pro-inflammatory, anti-tumor) or M2 (anti-inflammatory, pro-tumor) phenotypes is a highly dynamic process, influenced by a complex interplay of factors within the TME. The figure depicts the different signaling molecules—proteins/extracellular vesicles, colonic microbiota components/natural bioactive compounds, and ncRNAs (lncRNAs and miRNAs)—that play a crucial role in the regulation of macrophage polarization and activation. The cytokines responsible for M1 and M2 polarization are also shown. [(+)/(−)—positive/negative regulation; AI-2—*F. nucleatum* autoinducer-2; *A. muciniphila—Akkermansia muciniphila*; ARCR—*Astragalus mongholicus* Bunge-*Curcuma aromatica* Salisb; BBR—berberine; CB1—cannabinoid receptor 1; CPEB3—cytoplasmic polyadenylation element binding protein 3; Cp (Nem)—Cuban brown propolis (nemorosone); CRC-EVs—colorectal cancer-derived EVs; CTSK—metastasis-related secretory protein cathepsin K; *E. coli—Escherichia coli*; EGFR—epidermal growth factor receptor; EVs—extracellular vesicles; F1-MEV—F1 fraction of scorpion venom from *Mesobuthus eupeus*; *F. nucleatum—Fusobacterium nucleatum*; HMGA2—high mobility group A2; IFN-γ—interferon-gamma; IL-4/10/13—interleukin 4/10/13; lnc—long non-coding RNAs (e.g., NBR2, LINC00543, RPPH1, etc.); LPS—lipopolysaccharide; M0/1/2—macrophages of phenotype 0/1/2; M1EVs—M1 macrophage-derived EVs; MET—metformin; MFHAS1—malignant fibrous histiocytoma amplified sequence 1; miR—microRNAs (e.g., miR-18a, miR-148a, miR-21-5p, miR-216b, etc.); PCSK9—proprotein convertase subtilisin/kexin type 9; PKN2—protein kinase 2; PLXDC1—plexin domain containing 1; RNASET2—ribonuclease T2; TCF4—transcription factor 4; TNF-α—tumor necrosis factor-alpha].

**Figure 2 ijms-26-05336-f002:**
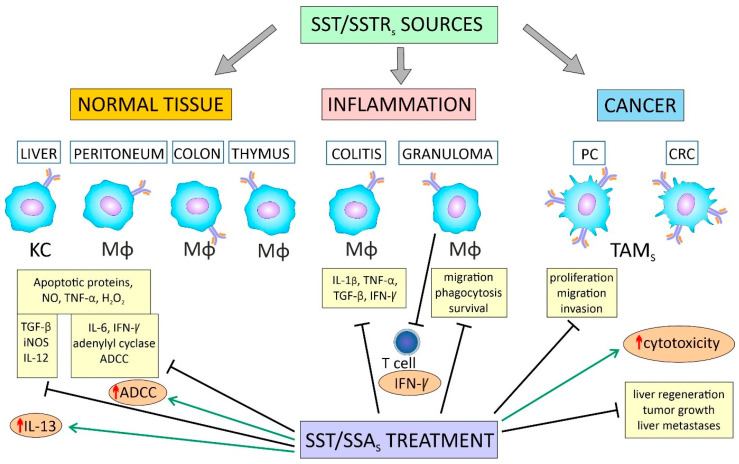
Macrophage population in homeostasis, inflammation, and cancers: the role of the SRIF system in macrophage regulation. TRMs and TAMs are both sources and targets of the SRIF system in health and disease. The impact of SST/SSAs involves preventing the production of pro-inflammatory cytokines (such as IL-1β, IL-6, TNF-α, and IFN-γ) and preventing the release of apoptosis proteins in both normal tissues and certain inflammatory conditions. In CRC, the SRIF system can both enhance macrophage cytotoxicity against cancer cells and directly inhibit tumor growth, but the mechanisms are only partially understood. [Arrows indicate positive regulation; ↑—increase; Ʇ—inhibition; ADCC—antibody-dependent cellular cytotoxicity; CRC—colorectal cancer; H_2_O_2_—hydrogen peroxide; IL-1β/6/12/13—interleukin 1β/6/12/13; IFN-γ—interferon-gamma; KC—Kupffer cell; Mφ—macrophage; (i)NO(S)—inducible (nitric oxide) synthase; PC—prostate cancer; SSA_s_—somatostatin analogs; STT—somatostatin; SSTR_s_—somatostatin receptors; SRIF—growth hormone-releasing inhibitory peptide/somatostatin; TAMs—tumor-associated macrophages; TGF-β—tumor growth factor beta; TNF-α—tumor necrosis factor-alpha; TRMs—tissue-resident macrophages].

**Table 1 ijms-26-05336-t001:** Functionally different proteins and extracellular vesicles (EVs) as regulators of TAMs in colorectal cancer.

Factor	Findings in CRC	Findings Concerning TAMs	Role in Carcinogenesis	Signaling Pathway	Ref. No.
EGFR	↑ (pEGFR, EGFR, iNOS) in AOM/DSS mice vs. NC	(i) ↑ M2 (Arg1, CCL17, CCL22, IL-10, IL-4) and ↓ M1 (iNOS, IL-12, TNF-α, CCR7) markers; (ii) IGF-1 from CRC cells—↑ M2 polarization; (iv) *Egfr* knockout prevents M1-to-M2 polarization	(i) ↑ Mφ-induced promotion of xenograft tumor growth; (ii) *Egfr* knockout—↓ cancer cell growth	EGFR	[98]
	↑ In colonic Mφ in the pre-cancerous stages of colitis and dysplasia	(i) ↓ M2 in *Egfr*-deficient mice—↓ Arg1/IL-10 mRNA and ↓ IL-4/IL-10/IL-13 proteins and ↓ M1; (ii) restrained M1/M2 activation—↓ CXCL1, VEGF, and ↓ CD31^+^ blood vessels	EGFR signaling in Mφ, but not in colonic epithelial cells, has a significant role in CAC		[99]
CB1	↓ CB1 and ↑ EGFR in tumor cells	↑ CB1 suppressed the differentiation of M2	↓ EGFR in CRC—↓ proliferation, migration, and invasion of CRC cells		[100]
PCSK9	(i) ↑ In tumor vs. NC; (ii) ♣ with advanced tumor grade	*PCSK9* knockdown—regulation of tumor EMT, ↓ M2 polarization, ↑ M1 polarization by ↓ lactate, protein lactylation, and MIF levels	*PCSK9* knockdown—↓ progression and meta of CRC	PI3K/AKT	[101]
MFHAS1	♣ Between expression in TAMs and TNM stage	CRC cells induce M2 polarization of TAMs through ↑ MFHAS1 and ↑ STAT6 and KLF4	↑ CRC progression	STAT6/KLF4	[102]
CPEB3	↓ In tumor ♣ with ↓ CD86+ and ↑ CD163+ TAMs	TAMs enhanced CRC cell proliferation and invasion via IL-6	Knockdown of *CPEB3*—↓ tumor progression and M2 polarization in vivo	IL-6R/STAT3	[103]
PLXDC1	(i) ↑ In the tumor; (ii) ♣ with shorter survival	(i) ♣ With M2 markers	Its downregulation may ↓ the progression of CRC		[104]
HMGA2	(i) ♣ With expression in tumor cells and CD68 in the stroma; (ii) ↑ CD68 predicts poor OS	↑ Mφ recruitment and M2 polarization in vitro and in vivo	↑ CRC progression	HMGA2/STAT3/CCL2	[105]
CTSK	↑ In tumor ♣ with meta and poor prognosis	(i) ↑ M2 TAMs in the stroma; (ii) bind to TLR4 to ↑ M2 polarization of TAMs	↑ Invasion and meta of CRC cells	NF-κB/mTOR	[106]
MET (AMPK agonist)	↑ (TNF-α and HLA-DR) and ↓ (Arg-1, CD163, CD206) in Mφ incubated with HCT116	Transforms TAMs to M1 Mφ	↓ progression of CRC	HIF-1α/p-AKT/mTOR	[107]
	↑ SCFA levels in ETBF/AOM/DSS mice	Modulates M2 polarization (inhibits M0-to-M2 transition)	↓ Colonic inflammation and colorectal tumorigenesis	TLR4/MyD88/NF-κB/MAPK	[108]
PKN2	(i) ↓ (IL4 and IL10) in tumor cells; (ii) predicts favorable prognosis and ♣ with ↓ M2 in human tumor	↓ M2 polarization both in vitro and in vivo	(i) ↓ Tumor growth in mice xenografts; (ii) ↓ tumorigenesis	DUSP6/ERK1/2	[109]
CD47	(i) ↑ In tumor; (ii) ↑ IL-10 secretion by tumor cells—↑ M2 Mφ polarization	(i) ↑ Mφ (CD68+) and ↑ M2 Mφ (CD206+); (ii) M2 Mφ secrete immunosuppressive cytokines	↑ Tumor cell migration and meta	CD47/SIRPα	[92]
RNASET2	↑ In cancer cells	(i) ↑ M1 recruitment and ↓ M2 Mφ; (ii) intra-tumor CD8+ T cells	↓ Tumor growth by modification of M1/M2 balance ratio, ↓ number of MDSCs, ↓ tumor vessels, and ↑ immune CD8+ T effector cells	N/A	[112]
TCF4	(i) ↑ In CRLM vs. primary tumor; (ii) controls CRC-derived CCL2	(i) ↑ TAM recruitment and M2 polarization in CRLM vs. primary CRC tissues; (ii) ♣ with ↑ TCF4 expression in meta	↑ Tumorigenesis and progression	TCF4-CCL2-CCR2	[113]
CRC-EVs	N/A	(i) SW480 EVs—↓ HLA-DR in M1 and M2, but ↑ CXCL10 in monocytes and M0; (ii) SW620 EVs—↑ (IL-6, CXCL10, IL-23, IL-10) in M0	Monocyte migration toward the TME and ↑ pro-inflammatory response by TAMs	N/A	[110]
M1EV1–4	N/A	TAMs shifted from M2 (CD163+) to M1 (CD68+) Mφ and ↓ (IL-10, TGF-β, CCL22)	M1EV2 and M1EV3 used alone or M1EV4—↓ tumor progression, ↓ primary tumor size and meta	STAT3/NF-κB/AKT	[111]

Legend: ↑/↓—high (upregulation/overexpression/promotion)/low (downregulation/lower expression; suppression); ♣—significant association; (p)AKT—(phosphorylated) protein kinase B; AMPK—adenosine monophosphate-activated protein kinase; AOM—azoxymethane; Arg-1—arginase-1; CAC—colitis-associated cancer; CB1—cannabinoid receptor 1; CCL2—C-C motif chemokine ligand 2; CCR2/7—C-C motif chemokine receptor type 2/7; CPEB3—cytoplasmic polyadenylation element binding protein 3; CRC—colorectal cancer; CRC-EVs—CRC-derived extracellular vesicles; CRLM—colorectal liver metastases; CTSK—metastasis-related secretory protein cathepsin K; CXCL1/10—C-X-C motif chemokine ligand 1/10; DSS—dextran sodium sulfate; DUSP6—dual specificity phosphatase 6; (p)EGFR—(phosphorylated) epidermal growth factor receptor; EMT—epithelial–mesenchymal transition; ERK1/2—extracellular signal-regulated kinase 1/2; ETBF—enterotoxigenic *Bacteroides fragilis*; EVs—extracellular vesicles; HLA-DR—human leukocyte antigen-DR; HIF-1α—hypoxia-inducible factor 1 alpha; HMGA2—high-mobility gene group A2; IGF-1—insulin-like growth factor; IL-4/6R/10—interleukin 4/6 Receptor/10/13; iNOS—inducible nitric oxide synthase; KLF4—Krüppel-like factor 4; LIPF1—non-PEGylated (HSPC/DSPG/Chol); LIPF2—PEGylated (HSPC/DSPG/Chol/mPEG2000-DSPE); M0/1/2—macrophages of phenotype 0/1/2; Mφ—macrophage; MAPK—mitogen-activated protein kinase; M1EV1—M1 macrophage EVs as nanocarriers of oxaliplatin; M1EV2—M1 macrophage EVs as nanocarriers associated with retinoic acid; M1EV3—M1 macrophage EVs as nanocarriers associated with *Libidibia ferrea*; M1EV4—M1 macrophage EVs alone or in combination; MDSCs—myeloid-derived suppressor cells; MET—metformin; meta—metastasis; MIF—macrophage migration inhibitory factor; Mφ—macrophage; Mφ-EVs—macrophage-derived extracellular vesicles; MFHAS1—malignant fibrous histiocytoma amplified sequence 1; mTOR—mammalian target of rapamycin; N/A—not applicable; NC—normal control; NF-κB—nuclear factor kappa-light-chain-enhancer of activated B cells; OS—overall survival; PCSK9—proprotein convertase subtilisin/kexin type 9; PI3K—phosphoinositide 3-kinase; PKN2—protein kinase 2; PLXDC1—plexin domain containing 1; Ref. No.—reference number; RNASET2—ribonuclease T2; SCFA—short-chain fatty acid; SIRPα—signal regulatory protein α; STAT3/6—signal transducer and activator of transcription 3/6; TAMs—tumor-associated macrophages; TCF4—transcription factor 4; TEVs—tumor-derived extracellular vesicles; TGF-β—transforming growth factor beta; TLR4—Toll-like receptor 4; TME—tumor microenvironment; TNF-α—tumor necrosis factor-alpha; TNM—tumor, node, metastasis; VEGF—vascular endothelial growth factor.

**Table 2 ijms-26-05336-t002:** Selected non-coding RNAs (ncRNAs) as epigenetic regulatory factors of TAMs in colorectal cancer.

Type of ncRNA	Findings in CRC	Findings Concerning TAMs	Role in Carcinogenesis	Signaling Pathway/Molecule	Ref. No.
lncRPPH1	(i) ↑ In tumor; (ii) ♣ with TNM and poor prognosis	TDEs mediate M2 polarization	↑ Meta and proliferation of CRC cells	↑ EMT/TUBB3	[118]
lncNBR2	(i) ↓ (NBR2, TNF-α, HLA-DR); (ii) ↑ (Arg-1, CD163, CD206, IL-4) in tumor	↑ M1 and ↓ M2 polarization in NBR2-overexpressed Mφ	↓ Progression of CRC in vitro and in vivo	N/A	[136]
	↓ NBR2 and ↑ miR-19a in tumor cells	(i) ↓ M2 polarization by ↓ miR-19a; (ii) NBR2 verified to target miR-19a in Mφ	↓ Progression of CRC by ↓ miR-19a	miR-19a	[137]
lncHLA-F-AS1	↑ In tumor	↑ PFN1 in CRC-EVs by ↓ miR-375—polarizing Mφ toward M2	↑ Tumorigenesis; ↑ migration, invasion, and EMT in vitro	miR-375	[129]
lncPTTG3P	(i) ↑ In tumor; (ii) ♣ with poor prognosis	Might ↑ cell proliferation, glycolysis through YAP1 and M2	↑ CRC progression	YAP1	[128]
lncMIR155HG	↑ In tumor	↑ M2 polarization	↑ CRC progression and drug resistance	miR-650/ANXA2	[130]
lncHOXB8-1:2	↑ In CgA Exo vs. NC Exo	(i) ↑ TAMs and M2 polarization; (ii) in TDEs, it acts as a ceRNA, competitively binding hsa-miR-6825-5p to ↑ CXCR3	↑ NE differentiated CRC progression	hsa-miR-6825-5p/CXCR3	[122]
lncHCG18	↑ in tumor	↑ M2 polarization to ↑ CET resistance	CET resistance	miR-365a-3p/FOXO1/CSF-1	[131]
LINC00543	(i) ↑ In tumor; (ii) ♣ with TNM stage and poorer prognosis	(i) ↓ Transport of pre-miR-506-3p across the XPO5—↓ miR-506-3p, resulting in ↑ FOXQ1 and ↑ EMT; (ii) ↑ FOXQ1—↑ CCL2 that ↑ Mφ recruitment and their M2 polarization	↑ CRC progression	pre-miR-506-3p/FOXQ1	[132]
lncXIST	↑ In THP-1 cells after treatment with CT26- and HCT116-derived Exo	↑ M2 Mφ in TDEs	↑ Proliferation, migration, and invasion of CRC cells	miR-17-5p/PDGFRA and AKT/ERK/STAT3/6	[124]
lncRP11-417E7.1	(i) ↑ In tumor with lymph node or distant meta; (ii) ♣ with a poor prognosis	(i) TDEs transport THBS2 into Mφ—↑ M2 polarization; (ii) binding with HMGA1—↑ THBS2 transcription	↑ CRC meta	Wnt/β-catenin/THBS2	[125]
lncBANCR	↑ In serum and tumor vs. adjacent tissues	TDEs (+) regulate the M2 polarization by intake IGF2BP2	↑ Cell proliferation, invasion, and meta	RhoA/Rock/IGFBP2	[126]
miR-21-5p and miR-200a	↑ In sEVs	Both ↑ M2 polarization and PD-L1 expression	↑ Tumor growth	PTEN/AKT and SCOS1/STAT1	[133]
miR-216b	↑ In tumor—↓ (CPEB4, CD206, IL-10)	(i) ↓ M2 polarization; (ii) targets CPEB4 to ↓ IL-10-mediated M2 polarization	↓ CRC development	N/A	[138]
miR-4766	↑ In hypoxia-treated Mφ	(i) ↓ M2 polarization; (ii) (−) regulates VEGFA	↓ CRC cell proliferation and migration	VEGFA	[139]
miR-I48a	(i) ♣ With M2 cytokines; (ii) ↑ expression—↓ Mφ recruited by SW480 cells	(i) ↑ M0-to-M1 differentiation and ↓ M2 polarization; (ii) ↓ TAM intake by targeting SIRPα	(i) ↓ CRC cell viability; anti-tumor effects	SIRPα	[140]
miR-106a-5p	↑ In Exo ♣ with CRLM and poor prognosis	(i) ↑ M2 polarization; (ii) hnRNPA1 regulates the transport into Exo	↑ CRLM	SOCS6/JAK2/STAT3/hnRNPA1	[127]
miR-122	↑ miR-122 and ↓ NEGR1 in CRLM vs. primary tumors	↓ NEGR1—↑ M2 polarization and ↑ CRLM in the mouse model	↑ CRC cell proliferation, migration, and invasion	PI3K/AKT	[134]
miR-18a	miR-18a encapsulated in GNV—↓ CRLM	miR-18a encapsulated in GNV—↑ M1 Mφ (F4/80+, IFN-γ+, IL-12+)	Anti-meta effect in CRC	N/A	[142]
circ-0034880	↑ In CRC-derived pEVs	TEVs-released circ-0034880—↑ SPP1^high^CD206+ pro-tumor Mφ	↑ PMN formation and CRLM	N/A	[135]
hUC-MSCs-Exos carrying miR-1827	(i) ↓ SUCNR1 in CRC—↓ tumor cell proliferation, migration, and invasion	↓ M2 Mφ polarization	(i) ↓ CRC cell growth; (ii) ↓ CRLM in vivo	N/A	[141]

Legend: (+)/(−)—positive/negative; ↑, ↓—high (upregulation/overexpression/promotion), low (downregulation/lower expression; suppression); ♣—significant association; AKT—serine/threonine kinase Akt, or protein kinase B (PKB); ANXA2—annexin A2; Arg-1—arginase-1; BANCR—BRAF-activated non-protein coding RNA; ceRNA—competitive endogenous RNA; CCL2—C-C motif chemokine ligand 2; CET—cetuximab; CgA—chromogranin A; CgA Exo—LoVo-CgA cells; CPEB4—cytoplasmic polyadenylation element binding protein 4; CSF-1—colony-stimulating factor 1; CXCR3—C-X-C motif chemokine receptor 3; CRC—colorectal cancer; CRLM—CRC liver metastasis; EMT—epithelial–mesenchymal transition; Exo(s)—exosome(s); ERK—extracellular signal-regulated kinase or classical MAP kinase (MAPK); EVs—extracellular vesicles; F4/80—adhesion G protein-coupled receptor E1; FOXO1—forkhead box O1; GNV—grapefruit-derived nanovector; HCG18—HLA complex group 18; HLA-DR—human leukocyte antigen-DR; HMGA1—high mobility group A1; hnRNPA1—heterogeneous nuclear ribonucleoprotein A1; HOXB8—homeobox B8; hUC-MSCs-Exos—exosomes from human umbilical cord mesenchymal stem cells; IGF2BP2—insulin-like growth factor binding protein 2; IFN-γ—interferon-gamma; IL-4/10/12—interleukin 4/10/12; JAK2—Janus kinase 2; lnc—long non-coding; M0/1/2—macrophages of phenotype 0/1/2; Mφ—macrophage; meta—metastasis; MIR155HG—MIR155 host gene; N/A—not applicable; NBR2—neighbor of BRCA1 lncRNA 2; NC Exo—LoVo-NC cells; NE—neuroendocrine; NEGR1—neuronal growth regulator 1; PDGFRA—platelet-derived growth factor receptor alpha; PD-L1—programmed death ligand 1; pEVs—plasma extracellular vesicles; PFN1—profilin 1; PI3K—phosphoinositide 3-kinase; PMN—pre-metastatic niche; PTEN—phosphatase and tensin homolog protein deleted on chromosome ten; PTTG3P—pituitary tumor-transforming 3 pseudogene; RhoA/Rock—Rho-associated protein kinase/a serine/threonine protein kinase; Ref. No.—reference number; RPPH1—ribonuclease P RNA component H1; SCOS1—suppressor of cytokine signaling-1; sEVs—small extracellular vesicles; SIRPα—signal regulatory protein α; SOCS6—SRY-box transcription factor 6; STAT1/3/6—signal transducer and activator of transcription 1/3/6; SUCNR1—succinate receptor 1; TAMs—tumor-associated macrophages; TDEs—tumor-derived exo; TEVs—tumor-derived extracellular vesicles; THBS2—thrombospondin 2; THP-1—Mφ cell line; TUBB3—β-III tubulin; VEGFA—vascular endothelial growth factor A; vs.—versus; YAP1—yes-associated protein 1; XPO5—exportin 5.

**Table 3 ijms-26-05336-t003:** Components of the colonic microbiota and natural bioactive compounds as regulators of TAMs in colorectal cancer.

Factor	Findings in CRC	Findings Concerning TAMs	Role in Carcinogenesis	Signaling Pathway/Molecule	Ref. No.
*A. muciniphila*	♣ With M1 Mφ and NLRP3/TLR2	↑ M1-like Mφ in vivo and in vitro	↓ CRC development	TLR2/NF-κB/NLRP3	[145]
AI-2 from *F. nucleatum*	↑ (TNFSF9 and IL-1β) in tumor vs. NC ♣ with AI-2 concentration and ↑ survival	↑ Mobility and M1 polarization in vitro	Anti-tumor effects	TNFSF9/TRAF1/p-AKT/IL-1β	[150]
	(i) ↑ TNFSF9 in tumor vs. NC; (ii) ♣ with CD3+ cells; (iii) ♣ with CD4/CD8 ratio in tumor	↑ TNFSF9 in Mφ of TME	Anti-tumor effects		[151]
*F. nucleatum*	(i) ↓ miR-1322 in CRC cells; (ii) ↑ tumor-derived CCL20	(i) ↑ Mφ infiltration through ↑ CCL20; (ii) ↑ M2 polarization	↑ CRC meta and TME reprogramming	miR-1322/NF-κB	[146]
	↑ Tumor-infiltrating M2 Mφ in *Fn*-positive tumors	(i) ↑ Differentiation M0 to M2 Mφ; (ii) ↑ M2 Mφ	↑ Cell proliferation, migration, and growth	TLR4/NF-κB/S100A9	[147]
*E. coli*	↑ In CRLM	(i) ↑ Lactate—↑ M2 polarization; (ii) RIG-I lactylation in M2 Mφ regulates PD-1+ Tregs and CD8+ T cells in the TME	↑ Tumor growth and CRLM	RIG-I/MAVS/NF-κB	[152]
ARCR	N/A	(i) ↓ infiltration of M2 Mφ into TME; (ii) ↓ ZFAS1 by ↓ Sp1 expression to ↓ M2 polarization	↓ CRC growth and meta in vivo	Sp1/ZFAS1/miR-153-3p/CCR5	[148]
Cp (Nem)	↓ HT-29 cell growth, migration, and invasion	(i) ↓ Viability of M2 Mφ and ↓ MMP-9 activity; (ii) M2 produce soluble factors	↓ Tumor cell growth	N/A	[90]
F1 fraction of MEV	↓ Migration and proliferation of tumor cells	(i) ↑ M1-associated cytokines/markers; (ii) ↓ M2-related markers; (iii) ↑ Mφ phagocytic capacity	Anti-tumor effects in vitro	N/A	[153]
BBR	(i) Protects against colon shortening in CAC; (ii) ↓ miR-155-5p and ↑ SOCS1	↓ The percentage of M1 Mφ and ↓ (IL-1β, IL-6, TNF-α) levels	Anti-tumor effects in the CAC mouse model	miR-155-5p/SOCS1	[156]
	↑ M2 and ↓ M1 Mφ in the colon and abdomen of DSS-induced UC mice	(i) ↑ (IL-4, STAT6, Chil3) and ↓ (TNF-α, IFN-γ, NOS2) expression in vivo; (ii) ↑ phosphorylation of STAT6 in vivo and in vitro to polarize M2 Mφ	Anti-inflammatory and immune-modulating activities	IL-4/STAT6	[91]

Legend: ↑, ↓—high (upregulation/overexpression/promotion), low (downregulation/lower expression; suppression); ♣—significant association; AI-2—autoinducer-2; *A. muciniphila—Akkermansia muciniphila*; ARCR—*Astragalus mongholicus* Bunge-*Curcuma aromatica* Salisb.; BBR—berberine; CAC—colitis-associated cancer; CCL20—C-C motif chemokine ligand 20; CCR5—CC chemokine receptor type 5; Chil3—chitinase-like 3; Cp (Nem)—Cuban brown propolis (nemorosone); CRC—colorectal cancer; CRLM—CRC liver metastasis; DSS—dextran sodium sulfate; *E. coli—Escherichia coli*; *F. nucleatum* (*Fn*)*—Fusobacterium nucleatum*; IFN-γ—interferon-gamma; IL-1β/4/6—interleukin 1β/4/6; M1/2—macrophages of phenotype 1/2; Mφ—macrophage; MAVS—mitochondrial antiviral signaling protein; meta—metastasis; MEV—scorpion venom from *Mesobuthus eupeus*; MMP-9—matrix metalloproteinase 9; N/A—not applicable; NC—normal control; NF-κB—nuclear factor kappa-light-chain-enhancer of activated B cells; NLRP3—NOD-like receptor protein 3, pyrin domain containing 3; NOS2—nitric oxide synthase 2; p-AKT—phosphorylated serine-threonine protein kinase; PD-1—programmed cell death protein 1; RIG-I—retinoic acid-inducible gene 1; SOCS1—suppressor of cytokine signaling 1; STAT6—signal transducer and activator of transcription 6; TAMs—tumor-associated macrophages; TLR2/4—Toll-like receptor 2/4; TME—tumor microenvironment; TNF-α—tumor necrosis factor-alpha; TNFSF9—TNF superfamily member 9; TRAF1—TNF receptor associated factor 1; UC—ulcerative colitis; ZFAS1—Zinc finger NFX1-type containing 1 anti-sense RNA 1.

**Table 4 ijms-26-05336-t004:** Prognostic value of TAM density in colorectal cancer tissues.

Material (No. of Cases) and Methods	Country	Findings	Prognostic Role	Year	Ref. No.
**↑ TAMs and Favorable Prognosis**					
pCRC (5); IHC (manually)	Singapore	(i) ↑ Pro-inflammatory TAMs; (ii) ♣ between tumor-infiltrating T cells and the no. of TAMs	↑ Mφ infiltration into CRC ♣ with good patient prognoses	2012	[75]
pCRC (485); IHC (manually using a 4-graded scale)	Sweden	(i) (+) ♣ Between the no. of NOS2+ (M1) and CD163+ (M2) cells; (ii) (−) ♣ to tumor stage for both NOS2+ and CD163+	↑ Infiltration by NOS2+ cells ♣ with better prognosis (CSS), independent of MSI and CIMP status	2012	[74]
pCRC (210); IHC (digital imaging scanning)	Bulgaria	(i) ↓ CD68+ cells infiltration in tumor stroma ♣ with expression of TGF-β1; (ii) ↓ CD68+ cells ♣ with LN meta, distant meta, ↑ stage, LVI, or perineural invasion	↓ CD68 appeared to be a significant unfavorable factor for prognosis	2013	[163]
pCRC (205); IHC in next-generation TMA (manually)	Greece	(i) ↑ CD68+ TAMs predicts less tumor budding; (ii) ↑ CD163+ TAMs indicative for < tumor grade; (iii) ↑ CD68+ and CD163+ TAMs ♣ with absence of LN meta	↑ CD68 ♣ with ↑ OS independent of Mφ polarization, microlocalization, pTNM, and post-operative therapy	2015	[165]
pCRC (419); IHC (computer-automated method)	China	↑ CD68+TF Mφ ♣ with ↑ CD44v6, ↓ Snail TF, and ↓ tumor buds	↑ CD68 TF Mφ predicted long-term OS	2017	[158]
pCRC (298); IHC (digital imaging scanning)	Austria	(i) Tumor Mφ with altered phenotype and ↓ CD206+ Mφ vs. control mucosa; (ii) nearly all CD206+ Mφ co-expressed CD163	↓ CD206+ Mφ ♣ with a poor prognosis	2018	[77]
pCRC (150); IHC (computer-assisted analysis)	Portugal	(i) CD163+ Mφ mainly at TF, CD80+ Mφ almost exclusively in ANM; (ii) CD163+ Mφ > in stage II, CD80+ Mφ in T1 tumors	(i) In stage III—↑ CD68 and ↓ CD80/CD163 ratio ♣ with ↓ OS; (ii) a protective role of CD80+ Mφ regarding the risk for relapse	2019	[166]
pCRC (931); digital image analysis and machine learning	USA, Finland/Australia	(i) ↑ Intraepithelial and stromal M1 Mφ ♣ with ↑ lymphocytic reactions; (ii) M2 Mφ ♣ with no. of TILs; (iii) MSI-high tumors have ↑ Mφ infiltration adjacent to tumor cells, and ↑ M1-like polarization of stromal Mφ	(i) ↑ Stromal density of M2-like Mφ ♣ with worse CSS; (ii) ↑ M1/M2 density ratio in tumor stroma ♣ with better CCS	2021	[159]
pCRC (64); IHC (Image Pro Plus, and intensity score 0–4)	China	CD86 expression (−) ♣ with tumor differentiation, LN meta, TNM stage, while CD163 expression (+) ♣ with tumor differentiation and tumor size	↑ CD86+ and CD68+CD86+ TAMs as well as ↓ CD163+ and CD68+ CD163+ TAMs ♣ with favorable OS	2022	[80]
pCRC (150); IHC (manually)	Russia	M2 Mφ—role in limiting the meta-process by affecting vascular maturity and normalization	↑ CD206 TAMs (+) ♣ with recurrence-free interval duration	2022	[81]
pCRC-TMAs (1720); IHC multiplex fluorescence IHC (digital image analyses)	Norway	(i) ↑ TAM in MSI vs. MSS tumors; (ii) CD68+ TAMs as prognostic markers with clinicopathological and genetic markers	(i) ↑ TAMs ♣ with favorable 5-year RFS in stage I–III; (ii) TAMs ♣ with a good prognosis in pts with ↑ T-cells in tumors; (iii) ↑ TAMs ♣ with poorer prognosis in pts with ↓ T cells; (ii) no difference in the 5-year OS in stage IV according to the no. of CD68	2024	[167]
↑ **TAMs and Unfavorable Prognosis**					
pCRC (159); IHC (manually)	Finland	↓ Intratumoral Stabilin-1+ TAMs among a high no CD68+ Mφ ♣ with a ↓ no. of distant recurrences	↑ Stabilin-1+ peritumoral Mφ (+) ♣ with longer DFS at stages II and III, but at stage IV—(+) ♣ with ↓ DSS rate		[168]
CRC (89); IHC (manually using a 4-graded scale)	Poland	The relative risks of recurrence and CRD were > in pts with ↑ M2 Mφ within the tumor vs. pts with no infiltration	(i) ↑ TAMs in the tumor stroma ♣ with ↓ DFS and OS with the opposite tendency at TF; (ii) TAMs CD68+/iNOS- and Tregs CD8+/FoxP3+ in tumor stroma are (−) prognostic factors with a (+) ♣ between them	2017	[164]
IHC; ELISA; IHC (image analysis software system)	China	(i) Preoperative serum MR and CD163 levels ♣ with serum CEA, CA19-9 and CA72-4 concentrations in CRC pts; (ii) ↑ MR and CD163 in CRC vs. para-cancerous tissues	↑ MR and CD163 expression in serum ♣ with ↓ OS and shown as adverse prognostic factors		[169]
pCRC (81); IHC (manually using immunoreactive score)	China	(i) CD163+ TAMs at TF ♣ with EMT, mesenchymal CTC ratio, and ↓ prognosis; (ii) TAMs—↑ EMT to ↑ CRC migration, invasion, meta by JAK2/STAT3/miR-506-3p/FoxQ1 axis, which leads to the CCL2 production promoting Mφ recruitment	↑ CD163+ expression ♣ with ↓ OS rate by 30% and ↓ RFS by 20%	2019	[170]
pCRC (81); IHC (manually using immunoreactive score)	China	(i) level of CD163+/CD68+ ratio in TF > in TC; (ii) ↑ CD163+/CD68+_TF_ ratio ♣ with ↑ LVI, tumor invasion, and TNM stage; (iii) ↑ CD163+/CD68+_TF_ ratio ♣ with EMT program and CTCs counts	(i) ↑ CD163+/CD68+_TF_ ♣ with both ↓ RFS and ↓ OS; (ii) CD163+/CD68+_TF_ better prognosticator vs. CD68+_TF_ and CD163+_TF_; (iii) M2 Mφ secreted TGF-β—↑ EMT, growth, proliferation, and invasion of CRC cells	2019	[171]
CRC biopsies (1008); IHC; prognostic models based on the training cohort (359)	China	pts with well-to-moderate tumor differentiation, fewer numbers of LN meta, TNM stage I or II, or RC tended to have ↑ TANs, ↓ Treg, or TAMs in the three cohorts	(i) TANs, Tregs, TAMs (CD163+) ♣ with worse prognosis and are independent prognostic factors; (ii) pts with ↑ TANs or ↓ Tregs and TAMs had a ↑ DFS and OS, independent of chemotherapy	2019	[73]
stage II CRC (521/314); IHC in TMA (digital imaging scanning)	China	↑ CD206/CD68 ratio ≥ 0.77 ♣ with LVI and perineural invasion	(i) ↑ CD206/CD68 ratio ♣ with ↓ DFS rate by 40% and OS by 30%; (ii) CD206/CD68 ratio has better prognostic efficacy than CD68^+^ and CD206^+^ TAMs, and other clinicopathologic high-risk factors	2019	[173]
submucosal-CRC (87); IHC (computer image analyzer)	Japan	(i) ↑ M2 in TF ♣ with LVI, ↓ histological differentiation, and LN meta; (ii) ↓ M1 in TF ♣ with LN meta; (iii)M2/M1 ratio better predictor of the risk of LN meta vs. pan-, M1, or M2 Mφ at TF	A marker comprising the phenotype, no, and distribution of TAMs may serve as a potential predictor of meta, including LN meta	2021	[79]
pCRC (242); IHC (manually)	China	↓ M1 (NOS2, CXCL10, CD11c) at TF and TC, while ↑ M2 (CD163, CD206, CD115) mainly at TF	(i) ↓ M1 and ↑ M2 markers exhibited ↓ OS rate; (ii) treble markers combination (NOS2/CXCL10/CD11c or CD163/CD206/CD115) are better predictors than a single marker	2023	[160]
pCRC (10 with CRLM); IHC (image analysis software)	USA	larger tumors (>3.9 cm), ♣ with ↑ total CD68+ Mφ, CD68+CD163+CD206− and CD68+CD206+CD163− M2 Mφ subtypes	↑ CD68+MRP8-14+CD86- M1 Mφ at TC ♣ with poor OS vs. low densities of this subtype		[96]

Legend: ↑/↓—increase/(overexpression)/decrease; (−)/(+)—negative/positive; >/<—higher/lower; ♣—significant association; ANM—adjacent normal mucosa; CA19-9/72-4—cancer (carbohydrate) antigen 19-9/72-4; CCL2—C-C motif chemokine ligand 2; CEA—carcinoembryonic antigen; CIMP—CpG island methylator phenotype; (p)CRC—(primary) colorectal cancer; CRD—cancer-related death; CSS—cancer-specific survival; CTCs—circulating tumor cells; CXCL10—C-X-C motif chemokine ligand 10; DFS—disease-free survival; DSS—disease-specific survival; ELISA—enzyme-linked immunosorbent assay; EMT—epithelial–mesenchymal transition; IHC—immunohistochemistry; JAK2—Janus kinase 2; LN—lymph node; LVI—lymphovascular invasion; meta—metastases/metastatic; M1/M2—phenotypes of macrophages; Mφ—macrophage; MR—mannose receptor; MRP8-14—myeloid-related protein 8-14; MSI—microsatellite instability; MSS—microsatellite stable; no.—number; NOS2 (iNOS)—nitric oxide synthase 2; OS—overall survival; pTNM—pathological tumor/node/metastasis; pts—patients; RC—rectum cancer; Ref. No—reference numbers; RFS—recurrence-free survival; STAT3—signal transducer and activator of transcription 3; TAMs—tumor-associated macrophages; TANs—tumor-associated neutrophils; TC—tumor center; TF—tumor invasive front; TGF-β—transforming growth factor beta; TILs—tumor-infiltrated lymphocytes; TMA—tissue microarray; Treg—regulatory T cells; vs.—versus.

**Table 5 ijms-26-05336-t005:** Summary of surface/cellular markers and cytokines produced by M1 and M2 subsets of TRMs in normal colon/rectum and TAMs in colorectal cancer.

Phenotype of TRMs/TAMs	Normal Colon/Rectum [Ref. No.]	CRC [Ref. No.]
M1	CD68/MHC-II/CD74 [65]	iNOS2/CD68 [76,80,158,163,164,165,166,171], CD86 [77,160,161,169], CD80 [166], CD64 [77,81], MHC-II [161], IL-12/CCR7/TNF-α [76], HLA-DR [77,81], co-expression of NOS2/CXCL10/CD11c [160], co-expression of CD68/CD80/MAF [82]
M2	CD68 [65]	CD68 [80,158,163,164,165,166,171], CD163 [74,76,80,164,165,169,170,171], TGF-β [163], Stabilin-1 [76,168], CD204 (MSR1) [76], CD206 (MRC1) [76,77,87,160,161,169], MDSC characteristic gene [87], MHC-II [161], co-expression of CD163/CD206/CD115 [160], co-expression of CD68/MARCO/VEGFA [82], CCL2/18/17/CXCL4 [82], IL-10/Arg1/CCL17/22/IL-4 [76]
Not specified	LN5/lysozyme/ferritin/α1-antichymotrypsin/25F9 [65], ACP5/C1Q, and LYVE1/COLEC12 [67], IL-4I1/FOLR2 [83]	VEGF [76], C1QC/SPP1 [82,83], IL-4I1/NLRP3 [83], mTORC2 [76]
Function	Antigen presentation and phagocytosis, regulation of genes related to immune activation and angiogenesis, regulation of genes related to neuronal homeostasis [67]; pathogen clearance, regulation of inflammatory responses, local tissue homeostasis, insulin sensitivity, and chronic inflammation [61]; epithelial self-renewal and metabolic support [68,70]	↓ Cell proliferation, ↓ production of cytokines (IL-6, IFN-γ) and chemokines (IL-8/CXCL8, CCL2) that attracted T cells and ↑ Th1 cell responses [75]; generation of a precancerous inflammatory environment facilitating the development of cancer [144]; release of pro-tumoral cytokines and chemokines and ↓ ability to activate T cells [161]; role in CAC initiation by ↑ production of cytokines (IL-1β, TNF-α, IL-6) which ↑ the stemness of Dclk1+ tufted cells [58]; ↑ cell proliferation, invasion, facilitating angiogenesis, ↑ tumor progression and metastases [7,79,88,171]; ↑ anti-tumor immune response [7,88]; regulation of metabolism [7]; interaction with microbiota [7,144]

Legend: ↑/↓—high (upregulation/overexpression/promotion)/low (downregulation/lower expression; suppression); 25F9—antigen on mature Mφ; ACP5—acid phosphatase 5; Arg1—arginase 1; *B. breve*—*Bifidobacterium breve*; CAC—colitis-associated cancer; C1Q—complement component 1Q; C1QC—complement C1q C chain; CCL2/18/17/22—C-C motif chemokine ligand 2/18/17/22; CCR7—C-C motif chemokine receptor 7; COLEC12—collectin subfamily member 12; CRC—colorectal cancer; CX3CR1—C-X3-C motif chemokine receptor 1; CXCL4/8/10—C-X-C motif chemokine ligand 4/8/10; Dclk1—doublecortin like kinase 1; FOLR2—folate receptor beta 2; HLA-DR—MHC class II cell surface receptor; iNOS2—inducible nitric oxide synthase 2; IL-1β/4/6/8/10/12—interleukin 1beta/4/6/8/10/12; IL-4I1—interleukin-4-induced gene 1; IFN-γ—interferon-gamma; LN5—laminin 5; LYVE1—lymphatic vessel endothelial hyaluronan receptor 1; Mφ—macrophage; MAF—musculoaponeurotic fibrosarcoma protooncogene; MARCO—macrophage receptor with collagenous structure; MDSC—myeloid-derived suppressor cell; MHC-II—major histocompatibility complex proteins class II; MRC1—mannose receptor C-type 1; MSR1—macrophage scavenger receptor 1; mTORC2—rapamycin-insensitive protein complex 2; NLRP3—NLR family pyrin domain containing 3; Ref. No.—reference numbers; SPP1—phosphoprotein 1; TAMs—tumor-associated macrophages; TGF-β—transforming growth factor beta; TNF-α—tumor necrosis factor-alpha; TRMs—tissue-resident macrophages; VEGFA—vascular endothelial growth factor A.

**Table 6 ijms-26-05336-t006:** TAMs as known and future targets for colorectal cancer therapy.

Positive Role	Drug [Ref. No.]	Promising Targets for Treatment [Ref. No.]		
		Proteins/EVs	ncRNAs	Natural Products
↑ monocyte/Mφ recruitment	CXCR4-CXCL12 inhibitors alone or with durvalumab (anti-PD-L1 Ab) [176]; CCR2/5 (BMS813160) with nivolumab (anti-PD1 Ab) or paclitaxel [62]	TCF4 [113]; HMGA2 in CRC and CAC [105]; CRC-EVs [110]	LINC00543 [132]	*F. nucleatum* [146]
↓ monocyte/TAM recruitment	NT157 (IGF-1R inhibitor) [179]; AMG 820 (anti-CSF-1R Ab) [176,197] or with pembrolizumab (CCR5 antagonist) [198]; pexidartinib (PLX3397, CSF-1R inhibitor) alone [176,199] or with durvalumab [176]; pembrolizumab [188]			
↑ M0 to M1 polarization		CRC-EVs [110]	miR-I48a [140]	
↑ M2 to M1 polarization (reprogramming TAMs)	Maraviroc [177,201]; CET [200]; regorafenib [202]; anti-MARCO IgG [189]; tasquinimod (S100A9 inhibitor) [190]; EZH2 inhibitors [191]; CT-0508 [177]; LIP-F1 and LIP-F2 [183]; pexidartinib alone [177] or with oncolytic viruses, and anti-PD-1 Ab [199]	M1EVs [111]		
↑ M1 polarization		EGFR in CAC [99]; RNASET2 [112]; MET [107]	miR-18a [142]; lncNBR2 [136,137]	*A. muciniphila* [145]; AI-2 from *F. nucleatum* [150]; F1 fraction of MEV [153]
↑ M2 polarization		EGFR in CRC and in CAC [98,99]; HMGA2 in CRC and CAC [105]; CB1 [100]; PCSK9 [101]; MFHAS1 [102]; CPEB3 [103]; PLXDC1 [104]; CTSK [106]; CD47 [92]; TCF4 [113]	lncRPPH1 [118]; lncPTTG3P [128]; lncHLA-F-AS1 [129]; lncMIR155HG [130]; lncHOXB8-1:2 [122]; lncHCG18 [131]; lncXIST [124]; lncRP11-417E7.1 [125]; lncBANCR [126]; LINC00543 [132]; miR-21-5p and miR-200a [133]; miR-106a-5p [127]; miR-122 [134]; circ-0034880 [135]	*F. nucleatum* [146,147]; *E. coli* [152]; BBR in CAC [91]
↓ M1 formation/polarization		EGFR in CRC and in CAC [98]; PCSK9 [101]		BBR in CAC [91,156]
↓ M2 formation/polarization		PKN2 [109]; RNASET2 [112]; MET [107]	lncNBR2 [136,137]; miR-216b [138]; miR-4766 [139]; miR-I48a [140]; hUC-MSCs-Exos carrying miR-1827 [141]	ARCR [148]; Cp (Nem) [90]; F1 fraction of MEV [153]
↓ M0 to M2 transition		MET in CAC [108]		
mixed M1/M2 cytokine response		mCRC-EVs [110]		
↓ TAM survival (depleting/reducing TAM numbers)	BPR1K871 [180]; pexidartinib (PLX3397) [192]; trifluridine/tipiracil (FTD/TPI) [193]; OXA [193]; M2pep [194]; RG7155 (anti-CSF-1R Ab) [195]; lenvatinib [196]			
↑ phagocytic activity of Mφ	Hu5F9-G4 with CET [175,176]; hAB21 alone or with PD-L1 inhibitors [185]; SIRP-1/SIRP-2 Ab, anti-CD47(AO-176) [186]			

Legend: ↑, ↓—high (upregulation), low (downregulation); Ab—antibody; AI-2—autoinducer-2; *A. muciniphila—Akkermansia muciniphila*; ARCR—*Astragalus mongholicus Bunge-Curcuma aromatica Salisb*.; BBR—berberine; CAC—colitis-associated cancer; CB1—cannabinoid receptor 1; CCR2/5—C-C motif chemokine receptor 2/5; CET—cetuximab; CPEB3—cytoplasmic polyadenylation element binding protein 3; CRC-EVs—CRC-derived extracellular vesicles; Cp (Nem)—Cuban brown propolis (nemorosone); CSF-1R—colony-stimulating factor 1 receptor; CTSK—metastasis-related secretory protein cathepsin K; CXCL12—C-X-C Motif Chemokine Ligand 12; CXCR4—C-X-C motif chemokine receptor 4; *E. coli—Escherichia coli*; EGFR—epidermal growth factor receptor; EZH2—enhancer of zeste 2 polycomb repressive complex 2 subunit; HMGA2—high-mobility gene group A2; hUC-MSCs-Exos—human umbilical cord mesenchymal stem cells; IGF-1R—insulin-like growth factor receptor; LIP-F1—non-PEGylated (HSPC/DSPG/Chol); LIP-F2—PEGylated (HSPC/DSPG/Chol/mPEG2000-DSPE); M1EVs—M1 macrophage-derived extracellular vesicles; M0/1/2—macrophages of phenotype 0/1/2; Mφ—macrophage; (m)CRC—(metastatic) colorectal cancer; MET—metformin; MEV—scorpion venom from *Mesobuthus eupeus*; MFHAS1—malignant fibrous histiocytoma amplified sequence 1; OXA—oxaliplatin; PCSK9—proprotein convertase subtilisin/kexin type 9; PD1/PD-L1—programmed cell death protein 1/ligand 1; PKN2—protein kinase 2; PLXDC1—plexin domain containing 1; Ref. No—reference numbers; RNASET2—ribonuclease T2; SIRP-1/2—signal regulatory protein α Ab; TAM(s)—tumor-associated macrophage(s); TCF4—transcription factor 4.

## Data Availability

No new data were created or analyzed in this study.

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
