# Peer review of "Regulation and Function of Tumor-Associated Macrophages (TAMs) in Colorectal Cancer (CRC): The Role of the SRIF System in Macrophage Regulation"

_ijms, 2025, doi:10.3390/ijms26115336_

Round 1
Reviewer 1 Report
Comments and Suggestions for Authors
- the title of the manuscript should be revised. It is not clear and it seems to be two titles not for one.
- the whole paper is a bit tedious, the authors should focus on what you want to explain,not all aspects of the field.
- the paragraph from line 54 to line 59 seems to be no relationship with macrophage., you should mention what is the association with TAM of those factors you would like to explain.
- the sentences should be rewrited from line 170 to line 173.
- It is better to add a pic about the regulation factors in polirization of macrophage in CRC, but not the types of macrophages.
- 4.1.1 should be reorganized, the whole part is disorderly, the logic is not clear, and you should put the central sentences at the first in the paragraph.
- for 5.1, I suggest to divide into in vitro, animal models, in viro three parts.
- the title for 7, is not appropriate, it has no relationship with macrophage or CRC,should be revised.
- 7.5 the title should be only one.
- Besides, there should be a transition betwween the macrophage and the SRIF, in manuscript it seems they are seperated 2 parts.
- the view point of the paragraph is not very clear, and the context shoul have transitions.
I think the quality of English is still needed to be polished, and some sentences are not very fluent and hard to read and understand. They often use too many "other" and "another" from line 292 to line 320. The
Author Response
Reviewer 1
Dear Reviewer,
I wish to thank you very much for a review, and time spent on reviewing the manuscript. Thank you very much for all your valuable comments and suggestions to revise my work to enhance the merit of the paper.
As recommended, I would like to address each of them:
Comments and Suggestions for Authors
the title of the manuscript should be revised. It is not clear and it seems to be two titles not for one.
As recommended by the reviewer, the title of the paper has been modified.
- the whole paper is a bit tedious, the authors should focus on what you want to explain,not all aspects of the field.
We divided the work into three parts, and the following sections were created according to this. We wanted to focus on the role of macrophages in CRC, but with an explanation of the potential role of the SRIF system on the function of macrophages as components of the immune system in general and the components of the TME in CRC.
- the paragraph from line 54 to line 59 seems to be no relationship with macrophage., you should mention what is the association with TAM of those factors you would like to explain.
We have shortened this paragraph and linked it to macrophages.
- the sentences should be rewrited from line 170 to line 173.
As suggested, the sentences in these lines have been rewritten.
- It is better to add a pic about the regulation factors in polirization of macrophage in CRC, but not the types of macrophages.
Our Figure 1 shows the involvement of the SRIF system in the regulation of macrophage function in „normal” macrophages and those present in examples of chronic inflammation (colitis, granuloma), and cancer (prostate cancer, CRC). We will not pass it up, as it is a summary of part three of our work.
However, at the reviewer's suggestion, an additional figure (currently Figure 1) was added, which presents a list of factors regulating the functions of macraphages in CRC (presented also in our Tables).
- 4.1.1 should be reorganized, the whole part is disorderly, the logic is not clear, and you should put the central sentences at the first in the paragraph.
As suggested by the reviewer, subsection 4.1.1. has been revised (also as to style and grammar), with clarification of unclear paragraphs. This entire subsection with Tables deals with new factors currently known to regulate macrophage polarization in CRC.
- for 5.1, I suggest to divide into in vitro, animal models, in viro three parts.
As suggested by the reviewer, we reorganized subsection 5.1. with a division into in vitro, animal models and in vivo studies.
- the title for 7, is not appropriate, it has no relationship with macrophage or CRC,should be revised.
As suggested by the reviewer, the title of section 7 and 7.5 have been modified to more closely define the role of the SRIF system in macrophage function in the healthy immune system and in the altered TME conditions in CRC. The next subsections of Chapter 7, namely 7.1.1, 7.2.1, 7.3, 7.5, and Figure 1, deal with macrophages in healthy tissues, in colitis, and in CRC. This section, although the findings are not spectacular, is the “novelty” of the entire review.
- 7.5 the title should be only one.
As suggested by the reviewer, the title of section 7.5 have been modified.
- Besides, there should be a transition betwween the macrophage and the SRIF, in manuscript it seems they are seperated 2 parts. the view point of the paragraph is not very clear, and the context shoul have transitions.
As suggested by the reviewer, some sentences linking this chapter to the entire work have been added.
All changes (and all additions) in the text were marked red. Thanking you once again for your efforts to read the work, valid comments, we ask you to be gracious to our efforts and positively accept the changes made.
I think the quality of English is still needed to be polished, and some sentences are not very fluent and hard to read and understand. They often use too many "other" and "another" from line 292 to line 320. The
The manuscript was once again read carefully, concerning the English language, and was proofread once again by a person who is certified in English and is a co-author of the paper (A.G.). We also used the English language correction programs generally available on the Internet (Grammar Checker & Rephraser, among others - https://www.grammarcheck.net/editor/).

Reviewer 2 Report
Comments and Suggestions for Authors
General comment
I read with great attention the manuscript entitled "Regulation and Function of Tumor-associated Macrophages 2 (TAMs) in Colorectal Cancer (CRC). Role of the SRIF System in 3 Macrophage Regulation in Normal Colon and CRC” by Geltz et al
The authors present a review that focuses on the role of tumor-associated macrophages (TAMs) in colorectal cancer (CRC), highlighting their dual pro-inflammatory and anti-inflammatory phenotypes. The review also addresses the lesser-studied role of the somatostatin (SRIF) system, which is known for its immunosuppressive effects, in modulating macrophage activity in CRC. It explores potential strategies for targeting TAMs and SRIF signaling pathways in CRC treatment. The manuscript is well-organized.
Additionally, the following observations are made to provide a better understanding of the study for readers of this journal.
- The English language used in the paper requires improvement in terms of sentence structure and grammar. A reformulation of certain sections would enhance overall clarity. Some minor typos also need to be corrected: One example: line 53 ”Consensus Molecular Subtype 1 (CMS1) and CSM4”, CSM4 should be CMS4.
The phrase “Both types are characterised by polymorphism.” It should be rewritten for clarification.
-The abstract is clear and provides a good summary of the main points of the study. However, the authors can restructure some sentences for better readability (e.g., "tumor-promoting activity" might be more explicitly tied to M2 macrophages). Also, the authors may give specific examples of signaling pathways or therapeutic strategies, which could strengthen their claims.
-In line 151, the authors point out that mouse studies have demonstrated the phenotypic heterogeneity of TRMs, some references should be introduced here.
- Table 5 should be organized like the other tables for better understanding.
- Due to the high number of abbreviations in the text, the authors should use the full name the first time it appears followed by the abbreviation, and in the cases where only once the abbreviation is used the full name should be instead (eg line 915 the phrase “In human GALT of the colon, SSTR expression was found mainly in secondary follicles” the use of “gut-associated lymphatic tissue” instead of “GALT” would make it more easy to understand).
Author Response
Reviewer 2:
Dear Reviewer,
I wish to thank you very much for a review, and time spent on reviewing the manuscript. Thank you and I really appreciate such a favorable review of our work.
I read with great attention the manuscript entitled "Regulation and Function of Tumor-associated Macrophages 2 (TAMs) in Colorectal Cancer (CRC). Role of the SRIF System in 3 Macrophage Regulation in Normal Colon and CRC” by Geltz et al
The authors present a review that focuses on the role of tumor-associated macrophages (TAMs) in colorectal cancer (CRC), highlighting their dual pro-inflammatory and anti-inflammatory phenotypes. The review also addresses the lesser-studied role of the somatostatin (SRIF) system, which is known for its immunosuppressive effects, in modulating macrophage activity in CRC. It explores potential strategies for targeting TAMs and SRIF signaling pathways in CRC treatment. The manuscript is well-organized.
Additionally, the following observations are made to provide a better understanding of the study for readers of this journal.
- The English language used in the paper requires improvement in terms of sentence structure and grammar. A reformulation of certain sections would enhance overall clarity. Some minor typos also need to be corrected: One example: line 53 ”Consensus Molecular Subtype 1 (CMS1) and CSM4”, CSM4 should be CMS4.
In accordance with the reviewer's remark, we tried to improve some parts of the work from the linguistic point of view (style, grammar) and remove the editorial errors noted. The noted language errors have also been corrected, and unnecessary abbreviations have been removed. The entire section 4.1.1. was rewritten, as well as some unclear parts of the work, including the abstract.
The manuscript was once again read carefully, concerning the English language, and was proofread once again by a person who is certified in English and is a co-author of the paper (A.G.). We also used the English language correction programs generally available on the Internet (Grammar Checker & Rephraser, among others - https://www.grammarcheck.net/editor/).
The phrase “Both types are characterised by polymorphism.” It should be rewritten for clarification.
The noted phrase has been removed, thank you for this comment.
-The abstract is clear and provides a good summary of the main points of the study. However, the authors can restructure some sentences for better readability (e.g., "tumor-promoting activity" might be more explicitly tied to M2 macrophages). Also, the authors may give specific examples of signaling pathways or therapeutic strategies, which could strengthen their claims.
Thank you for this comment, we have changed the suggested sentence structure and added the names of a couple of signal paths related to the review topic. However, the length limitations of the abstract do not allow for major changes.
-In line 151, the authors point out that mouse studies have demonstrated the phenotypic heterogeneity of TRMs, some references should be introduced here.
Thank you for this comment, this line has been rewritten due to ambiguities. Suggested references, regarding specific citations, have also been completed.
- Table 5 should be organized like the other tables for better understanding.
Indeed, Table 5 needed to be standardized according to the accepted scheme for other Tables. It was corrected to better understand its contents.
- Due to the high number of abbreviations in the text, the authors should use the full name the first time it appears followed by the abbreviation, and in the cases where only once the abbreviation is used the full name should be instead (eg line 915 the phrase “In human GALT of the colon, SSTR expression was found mainly in secondary follicles” the use of “gut-associated lymphatic tissue” instead of “GALT” would make it more easy to understand).
Thank you and for this comment, this has been corrected, and we have also tried to do this in other abbreviations as well. All changes (and all additions) in the text were marked blue. Thanking you once again for your efforts to read the work, valid comments, I ask you to be gracious to our efforts and positively accept the changes made.

Round 2
Reviewer 1 Report
Comments and Suggestions for Authors
I think it is ok for the manuscript and I don't have more questions.